# Mismatch repair deficiency endows tumors with a unique mutation signature and sensitivity to DNA double-strand breaks

Hui Zhao[1,2†], Bernard Thienpont[1,2†], Betül Tuba Yesilyurt[1,2†], Matthieu Moisse[1,2†], Joke Reumers[1,2], Lieve Coenegrachts[3], Xavier Sagaert[4], Stefanie Schrauwen[3], Dominiek Smeets[1,2], Gert Matthijs[5], Stein Aerts[5], Jan Cools[5,6], Alex Metcalf[7], Amanda Spurdle[7], ANECS[8], Frederic Amant[3], Diether Lambrechts[1,2]*

[1]VIB Vesalius Research Center, KU Leuven, Leuven, Belgium; [2]Department of Oncology, KU Leuven, Leuven, Belgium; [3]Division of Gynaecologic Oncology, Department of Obstetrics and Gynaecology, University Hospital Gasthuisberg, Leuven, Belgium; [4]Division of Pathology, University Hospital Gasthuisberg, Leuven, Belgium; [5]Department of Human Genetics, KU Leuven, Leuven, Belgium; [6]VIB Center for the Biology of Disease, KU Leuven, Leuven, Belgium; [7]Division of Genetics and Computational Biology, Queensland Institute of Medical Research, Brisbane, Australia; [8]The Australian National Endometrial Cancer Study, PO Royal Brisbane Hospital, Brisbane, Australia

**Abstract** DNA replication errors that persist as mismatch mutations make up the molecular fingerprint of mismatch repair (MMR)-deficient tumors and convey them with resistance to standard therapy. Using whole-genome and whole-exome sequencing, we here confirm an MMR-deficient mutation signature that is distinct from other tumor genomes, but surprisingly similar to germ-line DNA, indicating that a substantial fraction of human genetic variation arises through mutations escaping MMR. Moreover, we identify a large set of recurrent indels that may serve to detect microsatellite instability (MSI). Indeed, using endometrial tumors with immunohistochemically proven MMR deficiency, we optimize a novel marker set capable of detecting MSI and show it to have greater specificity and selectivity than standard MSI tests. Additionally, we show that recurrent indels are enriched for the 'DNA double-strand break repair by homologous recombination' pathway. Consequently, DSB repair is reduced in MMR-deficient tumors, triggering a dose-dependent sensitivity of MMR-deficient tumor cultures to DSB inducers.

*For correspondence: diether. lambrechts@vib-kuleuven.be

†These authors contributed equally to this work

## Introduction

MMR-deficiency represents a well-established cause of Lynch syndrome, which is an autosomal dominantly inherited disorder of cancer susceptibility triggered by loss-of-function mutations in MMR genes (*MLH1*, *MSH2*, or *MSH6*) (*Jiricny, 2006*). Lynch syndrome is responsible for 2–5% of endometrial (EM) or colorectal (CRC) tumors. Additionally, epigenetic silencing of *MLH1* contributes to another 15–28% of these tumors (*Parsons et al., 2012*; *Peltomaki, 2014*). Deficiency of the MMR machinery leads to DNA replication errors in the tumor tissue, but not the normal surrounding tissue. In particular, errors often accumulate as indel mutations in mono- and di-nucleotide repeats—a phenomenon referred to as microsatellite instability (MSI) (*Pinol et al., 2005*).

MMR-deficient tumors exhibit a different prognosis and therapeutic outcome after standard chemotherapy (*Ng and Schrag, 2010*). Untreated CRC patients with MMR-deficient tumors have a modestly

**eLife digest** Before a cell divides, it must first copy all of its genetic material. Any mistakes that are made during this process are called mutations. Mutations can give rise to new traits but are mostly harmful to the cells, or cause cancer; therefore, cells have evolved tools that can efficiently spot these mistakes and repair them. One of the main tools is called mismatch repair (MMR).

Defects in the cell's mismatch repair tools can wreak havoc as this allows many mutations to accumulate. Zhao et al. looked at the genomes of tumors where mismatch repair was not working properly to see what makes these 'MMR-deficient tumors' different from other tumors. This revealed that MMR-deficient tumors have similar patterns of mutations to those seen in egg and sperm cells. This was unexpected and suggests that mutations that are not corrected by mismatch repair are an important source of the genetic differences found between different humans, and between humans and their ancestors.

Identifying cancerous tumors that are MMR-deficient is vital, as these tumors tend not to respond to commonly used cancer treatments. However, current clinical methods to identify MMR-deficient tumors often fail or produce results that are difficult to interpret. MMR-deficient tumors commonly contain mutations called indels, where short fragments of DNA are inserted or deleted into longer DNA sequences. Zhao et al. have found 59 indels that can be used to detect MMR-deficient tumors, where each indel had been identified in several tumors taken from different tissues. This new approach allowed MMR-deficiency to be identified in several types of tumor, including colon and ovarian cancers, with greater sensitivity and accuracy than the existing methods.

Zhao et al. also found that the indels in MMR-deficient tumors reduce the ability of the tumors to repair a type of DNA damage called double-strand breaks. In these, both strands of DNA that make up the double helix are broken and the DNA chain is severed. As this kind of damage is very harmful to a cell, making more double-strand breaks could therefore form part of a more effective treatment against MMR-deficient tumors; further research is needed to investigate this possibility.

better prognosis, but do not seem to benefit from 5-fluorouracil-based adjuvant chemotherapy, which is the first-choice chemotherapy for CRC. In particular, in MMR-deficient tumors, mismatches induced by 5-fluorouracil are tolerated, leading to failure to induce cell death (*Fischer et al., 2007*). MMR-deficient tumors are also resistant to cisplatin and carboplatin, which are frequently used chemotherapies in EM cancer (*Hewish et al., 2010*). Furthermore, MMR-deficient tumors can be resistant to targeted therapies, because they acquire secondary mutations in genes that activate alternative or downstream signaling pathways (e.g., *PIK3CA*). Another possibility is that epigenetic silencing of *MLH1* coincides with particular mutations, such as the *BRAF* V600E mutation (*Donehower et al., 2013*), which represents an established negative predictor of response to targeted anti-EGFR therapies in advanced CRC (*Richman et al., 2009*).

Efforts to individualize the treatment of MMR-deficient tumors have focused on identifying synthetic lethal interactions within the MMR pathway. In particular, increased oxidative damage (by methotrexate exposure or *PINK1* silencing [*Martin et al., 2011*]) and interference with the base excision repair (BER) pathway (by DNA polymerase γ or β inhibition [*Martin et al., 2010*]) can sensitize MMR-deficient tumors. Until now, these findings failed, however, to translate into clinically effective treatment options. Alternatively, as highlighted above, secondary mutations occurring because of MMR-deficiency may also critically determine therapeutic efficacy (*Dorard et al., 2011*). These secondary mutation spectra have, however, been poorly characterized, mainly because studies often focused at one or a few reporter loci, or exclusively on mutations at known hotspot sequences. More recently, the first whole-exome sequencing of MMR-deficient tumors was performed, highlighting the clearly distinct mutational landscape of these tumors (*TCGA, 2012*), whereas at the whole-genome level, *Kim et al. (2013)* revealed overrepresentation of MSI in euchromatic and intronic regions compared to heterochromatic and intergenic regions.

To generate a more comprehensive picture of the mutation spectra arising in MMR-deficient tumors, and in particular, to interpret their clinical relevance with respect to diagnostically assessing MSI and therapeutically targeting MMR-deficient tumors, we sequenced another comprehensive set

of MMR-deficient tumors. In particular, whole-genome and whole-exome sequencing was applied to 5 and 28 tumor–normal pairs, of which respectively 3 and 22 were MMR-deficient.

## Results

### Whole-genome sequencing of MMR-deficient tumors

To select MMR-deficient tumors for whole-genome sequencing, standard diagnostic tests were used, including immunohistochemistry of MMR proteins (MLH1, MSH2, and MSH6), assessment of MSI using the extended Bethesda panel and methylation profiling of the *MLH1* promoter. Three chemo-naive EM tumors, either deficient for MLH1, MSH2, or MSH6 and thus covering the full spectrum of MMR-deficiency, as well as two MMR-proficient EM tumors were selected (*Table 1*). Different sequencing technologies were leveraged to avoid potential technology biases in assessing mutation patterns in MMR-deficient tumor genomes, that is, Complete Genomics (CG) and Illumina short-read sequencing. We obtained high coverage sequencing data (30–120x) for tumor and matched normal samples (*Table 1*). Application of a standard annotation and filtering pipeline, as previously described (*Reumers et al., 2011*), revealed that each MMR-deficient tumor exhibited a clear hypermutator phenotype, containing on average 50 times more novel somatic mutations than MMR-proficient tumors (*Figure 1A*, *Figure 1—source data 1*, *Figure 1—source data 2*). Orthogonal technologies validated 98% of substitutions and 88% of indels in the three MMR-deficient tumors, while only 62% of substitutions and 11% of indels were validated in the two MMR-proficient tumors (*Figure 1—source data 3*). This difference in validation rates between MMR-deficient and MMR-proficient tumors is probably due to the fact that in normal genomes, as well as MMR-proficient tumor genomes, the number of true-positive indels is low in comparison to the number of false-positive indels. However, in MMR-deficient tumors, due to their specific hypermutator phenotype, the number of true-positive indels is vastly increased, thereby rendering the false positive fraction proportionally much smaller. Notably, all tumors were negative for *POLE* mutations (*Kandoth et al., 2013*; *Palles et al., 2013*).

### Somatic mutation patterns in MMR-deficient hypermutators

Studies in model organisms and cell lines have shown that somatic mutations arising due to MMR-deficiency mostly involve indels affecting microsatellite sequences (di- to hexa-nucleotide repeats with a minimal length of six bases and at least two repeat units) and homopolymers (mononucleotide repeats with a minimal length of six bases) (*Ellegren, 2004*). We observed that indels were indeed more frequent than single basepair substitutions in all three MMR-deficient tumors (*Figure 1A*). Indels predominantly affected homopolymers (40-fold enrichment over expected by chance) and to a lesser extent also microsatellites (2.3-fold enrichment; *Figure 1B*, *Figure 1—figure supplement 1*). Substitutions were only slightly enriched in homopolymers and microsatellites (3- and 1.5-fold enrichment, respectively; *Figure 1B*). Mutations occurred as frequently in introns as in the rest of the genome, but were clearly less frequent in exons (excluding 5' and 3' untranslated regions [UTRs]). This decrease was caused by indels that were 91% less frequent in exons (*Figure 1C,D*). Correction for the number of homopolymers, the length of homopolymers or their basepair composition in exons versus

**Table 1.** Standard diagnostic tests to assess MMR-deficiency

| Tumor | Histopathology | Grade | Stage | Coverage | | IHC | | | MSI | *MLH1* hyper-methylation |
| | | | | Tumor | Germ-line | MLH1 | MSH2 | MSH6 | | |
|---|---|---|---|---|---|---|---|---|---|---|
| MMR– 1 | Endometrioid | 3 | IIIc | 87.1 | 81.1 | + | + | –(*) | + | – |
| MMR– 2 | Serous/clear cell | 3 | Ib | 24.8 | 21.9 | + | – | – | – | – |
| MMR– 3 | Endometrioid | 2 | Ib | 28.5 | 30.0 | – | + | + | + | + |
| MMR+ 1 | Endometrioid | 3 | I | 119.4 | 73.1 | + | + | + | – | + |
| MMR+ 2 | Serous | 3 | Ia | 79.2 | 77.0 | + | + | + | – | – |

Tumors and matched germ-line were whole-genome sequenced using either Complete Genomics or Illumina sequencing technology. For each tumor, microsatellite instability (MSI) using the extended Bethesda panel, standard immunohistochemistry of MMR proteins (MLH1, MSH2, and MSH6), and methylation status of the *MLH1* promoter are shown.

*a weak positive nuclear staining in the minority of the tumor cells.

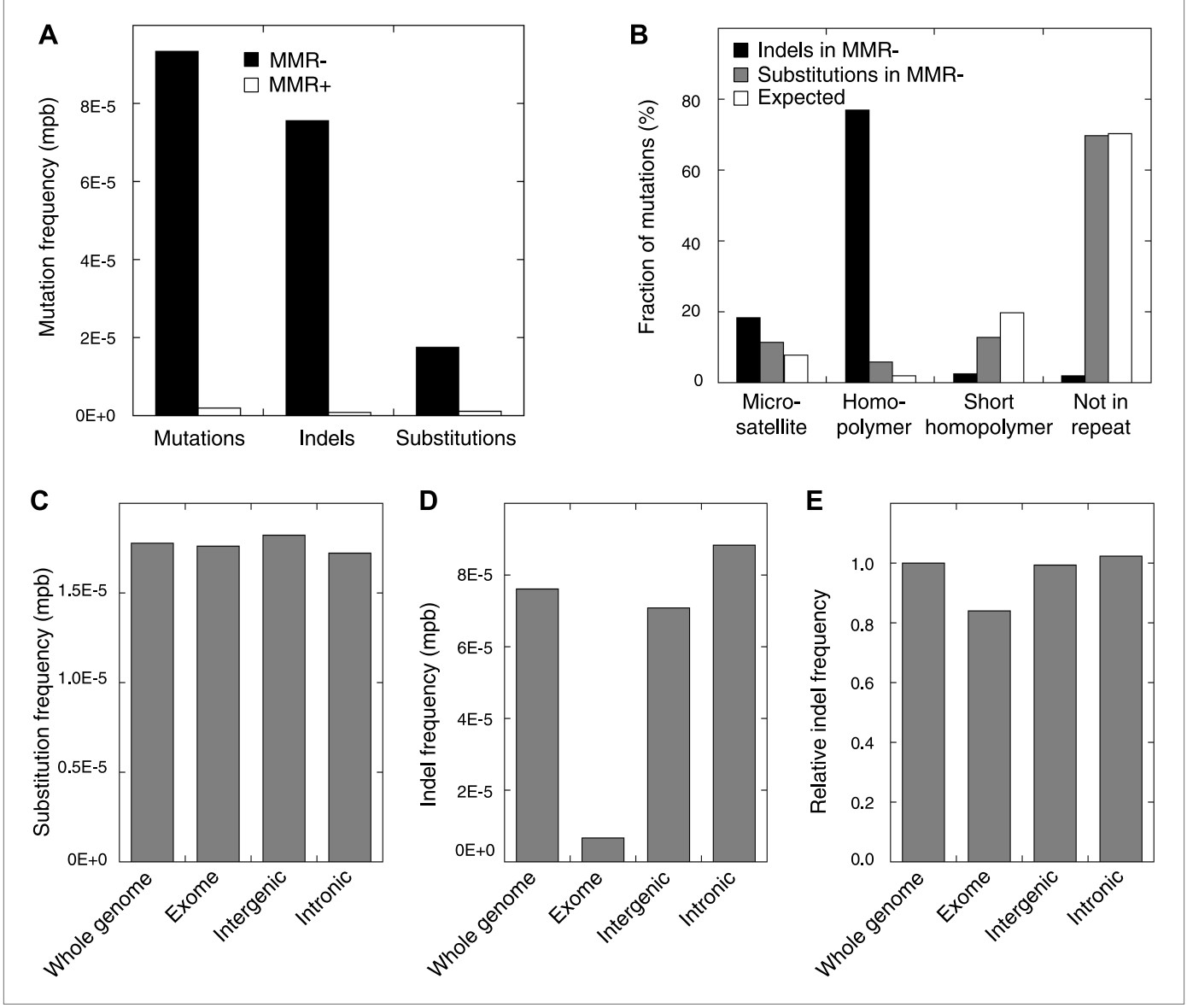

**Figure 1**. Somatic mutations in MMR-deficient tumors. (**A**) The average frequency of mutations, indels, and substitutions in MMR-deficient tumors vs MMR-proficient tumors, expressed as number of mutations per base (mpb). (**B**) The fraction of indels and substitutions observed in microsatellites, homopolymers (length over 5 bp), short homopolymers (length of 3–5 bp), and 'not in repeat regions' compared to their expected fraction in these regions. (**C** and **D**) Frequencies of substitutions (**C**) and indels (**D**) in MMR-deficient tumors stratified into exonic, intergenic, and intronic regions. (**E**) Indel frequencies corrected for homopolymer number, length, and base composition. Indel frequencies in MMR-deficient tumors represent estimates only, as orthogonal technologies revealed false-positive rates of 12%, while false-negative rates in CG and Illumina whole-genomes were estimated to be 27.7% and 0.5%, respectively, by *Zook et al. (2014)*. In MMR-proficient tumors all detected somatic indels were independently validated.

The following source data and figure supplements are available for figure 1:

**Source data 1**. Sequence statistics of MMR-proficient and MMR-deficient whole genome sequenced tumour samples, and a list of somatic substitutions detected therein.

**Source data 2**. List of somatic indels detected in the MMR-proficient and MMR-deficient, whole genome sequenced tumour samples.

**Source data 3**. List and overview of validated somatic mutations, detected in the MMR-proficient and MMR-deficient whole genome sequenced tumour samples.

*Figure 1. Continued on next page*

*Figure 1. Continued*

**Figure supplement 1**. The fraction of indels (left panel) and substitutions (right panel) observed in microsatellites, homopolymers, short homopolymers and in nonrepeat regions compared to their expected fraction in these regions.

**Figure supplement 2**. The relative indel frequency defined as the number of indels divided by the total bases of non-homopolymer regions in MMR-deficient tumors stratified into intergenic, exonic, 5'UTR, 3'UTR, and intronic regions is shown.

**Figure supplement 3**. Copy number status of the 5 whole-genomes assessed by Illumina Human-Omni1 and CytoSNP-12 chips.

other regions weakened this effect, but failed to completely alleviate it (*Figure 1E*, *Figure 1—figure supplement 2*). Since 92% of exonic indels resulted in frameshift mutations, which have a greater functional impact than substitutions (*Montgomery et al., 2013*), this suggests that exonic indels are prone to negative clonal selection during tumorigenesis.

## Somatic substitutions in MMR-deficient hypermutators

There is extraordinary variation in the frequency and spectrum of somatic mutations affecting different cancers, shedding light on the underlying mutational processes and disease etiology of these tumors (*Wheeler and Whang, 2013*). When assessing somatic substitutions in MMR-deficient tumors, we observed that 74% of all substitutions represent transitions (i.e., purine-to-purine or pyrimidine-to-pyrimidine substitutions), which is similar to the patterns observed in the matched germ-line of these tumors (*Figure 2A*). This is surprising, since tumor genomes generally display patterns distinct from those found in the germ-line. Indeed, when extending these analyses to other hypermutators, that is, UV-light-induced melanoma (*Pleasance et al., 2010*), tobacco smoke-induced small cell lung adenocarcinoma (SCLC) (*Pleasance et al., 2010*), as well as breast tumors deficient for BRCA1 (*Nik-Zainal et al., 2012*) or EM tumors proficient for MMR, patterns were clearly dissimilar from the matched germ-line (*Figure 2A*). On the other hand, de novo germ-line substitutions identified through whole-genome sequencing of parent–offspring trios (*Campbell et al., 2012*; *Kong et al., 2012*), common genetic variation as catalogued by the 1000 Genomes Project (1 KG) (*1000 Genomes Project Consortium, 2012*), and substitutions that occurred in the human lineage during the divergence of humans and chimpanzees correlated strongly to the MMR-deficient tumor genome (*Figure 2A*). Given these remarkable parallels, we hypothesized that MMR-deficient genomes hypermutate in a way that mirrors the processes driving genetic variation on a population level, albeit somatically and on a shorter time scale.

To further assess the similarities between MMR-deficient mutation patterns and germ-line genetic variability, we analyzed small-scale and large-scale context-dependent effects on substitution patterns. At the small-scale level, when assessing the effect of flanking nucleotides on substitution frequencies, the patterns of all four sets of germ-line genetic variants were highly correlated to MMR-deficient tumors (average $R^2 = 0.77$), but less to the four other cancer genomes (average $R^2 = 0.45$; *Figure 2B,C*), providing further support for our hypothesis. On a large-scale context, the number of intergenic substitutions per 1 Mb in germ-line genetic variability databases was similarly highly correlated to those in MMR-deficient genomes (average $R^2 = 0.67$), but not to those in other cancer genomes (average $R^2 = 0.42$; *Figure 2D*). This suggests that also on a large scale, substitutions are comparably distributed in MMR-deficient tumor genomes as in germ-line genomes. At the large-scale level, nine genomic features are linked with genetic variability (*Hodgkinson and Eyre-Walker, 2011*). Each of these features correlated significantly with substitution frequencies in MMR-deficient tumors and germ-line genomes. Linear modeling revealed that six of these independently correlated with substitution rates in MMR-deficient tumors as well as with germ-line substitutions (*Figure 2E*). Overall, the types as well as the narrow and broad context-dependencies of substitutions thus appear to be largely shared between germ-line and MMR-deficient genomes, suggesting that a considerable fraction of human genetic diversity arises through mismatches escaping MMR.

Since MMR-proficient tumors carried 50 times fewer substitutions and displayed more disparate substitution patterns than MMR-proficient tumors, the observed correlations can almost exclusively be attributed to the MMR-deficient phenotype of these tumors. As such, these correlations also provide novel insights into the functioning of the MMR system. First, replication timing correlated with transitions

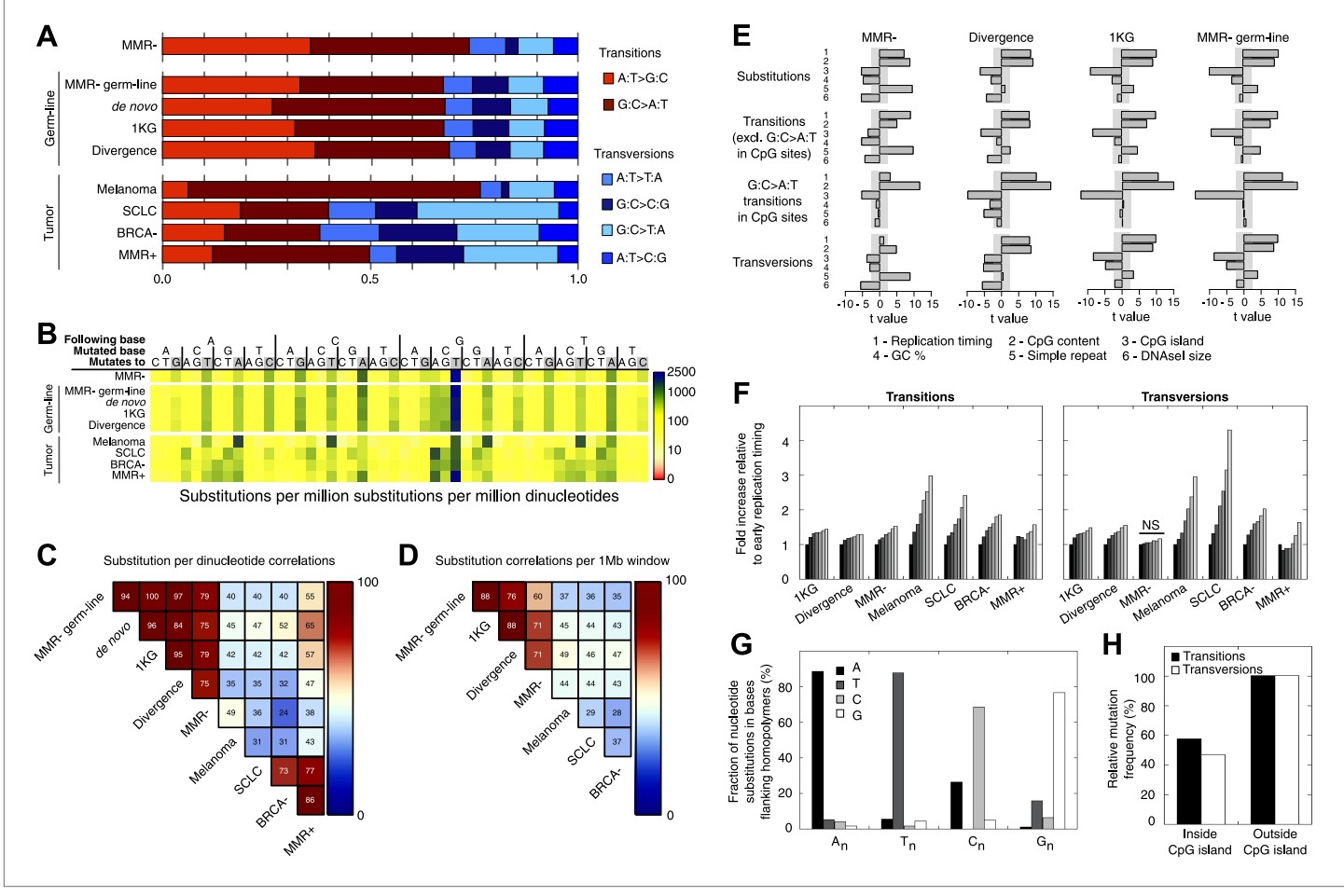

**Figure 2**. Somatic substitution patterns in MMR-deficient tumors. (**A**) Somatic substitution patterns in whole-genome sequences of MMR-deficient endometrial tumors (MMR−), matched germ-line (peripheral white blood cell) DNA from MMR-deficient tumors (MMR-germ-line), de novo mutations as identified in parent-offspring trios (de novo), 1000 Genomes Project (1 KG), the human–chimpanzee divergence panel (Divergence), melanoma and small-cell lung cancer (SCLC), BRCA-deficient breast tumors (BRCA−), MMR-proficient endometrial tumors (MMR+). (**B**) Somatic substitution frequency per million dinucleotides and per million substitutions. The first row lists the base following the mutated base, the second row lists the base that was mutated, and the third row lists the new base. Gray boxes indicate transitions. Frequencies are depicted color-coded following a logarithmic distribution as shown by the gradient on the left. (**C** and **D**) Squared coefficients of correlation ($R^2$) between dinucleotide substitution patterns (**C**) and between the number of intergenic substitutions per 1 Mb window (**D**). Substitutions in MMR-proficient and de novo data sets were too sparse for correlations at a 1 Mb scale. (**E**) Multivariate linear regression modeling of genomic features predicting substitutions frequencies per 1 Mb window in MMR-deficient tumors, and the outcome of the same multivariate linear regression modeling in the germ-line genetic variability panels. T-values resulting from the linear model are displayed as bar plots and indicate direction and significance of correlation (shaded grey box equals p > 0.05, Bonferroni-corrected per model). The de novo substitution frequency was too low to be modeled at this resolution. (**F**) Frequency of transitions (excluding G:C>A:T in CG) and transversions per 1 Mb window, binned per replication time. Frequencies are displayed relative to the earliest replicating bin. Linear regression analysis was performed to assess whether observed increases were significant and independent of other genomic features. All Bonferroni-corrected p-values were significant (p < 2.0E−5) except for transversions in MMR-deficient tumors, which were not significant (NS; p = 0.23). (**G**) Effect of homopolymer nucleotide composition ($A_n$, $T_n$, $C_n$, or $G_n$) on substitutions immediately flanking a homopolymer. For example, the nucleotide B next to the poly-A repeat 'NNB(A)$_n$BNN' is mostly converted to an A (NNB(A)$_n$ANN) and not to a C, G, or T. The modest increase in A substitutions next to $C_n$ homopolymers and T substitutions near $G_n$ homopolymers is caused by C:G>T:A transitions in a CpG context. (**H**) Substitution frequency in and outside CpG islands, relative to genome-wide substitution frequencies. Data combined for all three MMR-deficient genomes are represented for (**B**, **E–H**), but individual MMR-deficient genomes display similar patterns (**Figure 2—figure supplements 1–5**).

The following figure supplements are available for figure 2:

**Figure supplement 1**. Somatic substitution frequency per million dinucleotides and per million substitutions for the individual MMR-deficient genomes.

*Figure 2. Continued*

**Figure supplement 2**. Multivariate linear regression modeling of genome features predicting substitutions frequencies per 1 Mb window in the individual MMR-deficient genomes.

**Figure supplement 3**. Frequency of transitions (excluding G:C>A:T in CG) and transversions per 1 Mb window, binned per replication time, relative to the earliest replicating bin.

**Figure supplement 4**. Effect of homopolymer nucleotide composition ($A_n$, $T_n$, $C_n$, or $G_n$) on substitutions immediately flanking a homopolymer in the individual MMR-deficient genomes.

**Figure supplement 5**. Frequency of transitions and tranvsersions in and outside of CpG Islands in the individual MMR-deficient genomes.

but not transversions in all three MMR-deficient tumors (*Figure 2F*). This contrasts with the increase in late S phase transversions observed in all other genomes studied here (*Figure 2F*), as well as in lymphoblastoid cell lines (*Koren et al., 2012*). The increase in MMR-proficient but not MMR-deficient cells suggests a reduced fidelity of DNA repair in late S phase, leading to an increase in transversions. Potential causes include a decreased MMR-activity in late S phase, or a longer window of time available for the repair of early vs late transversions in MMR-proficient cells (*Hombauer et al., 2011*). In contrast, DNA repair fidelity in MMR-deficient cells is invariably low and therefore not affected by replication time. Secondly, a positive association with simple repeat content was noted. Indeed, a 1.6-fold increase in substitutions at bases immediately flanking simple repeats was noted, with a threefold increase next to homopolymers and a 1.3-fold increase next to microsatellites (*Figure 2G*). These substitutions for the vast majority converted the base flanking the repeat, to the base constituting the repeat (*Figure 2G*). They are thus probably the result of polymerase slippage events, following a mechanism akin to the previously described bacterial dislocation mutagenesis (*Kunkel and Soni, 1988*). Thirdly, G:C>A:T transitions in CpG sites strongly depend on CpG content, but are inversely correlated with the fraction of CpG islands (*Figure 2E*). Spontaneous, replication-independent deaminations of methyl-C to T underlie such transitions. Here, the much larger increase in CG>TG transitions observed in MMR-deficient compared to MMR-proficient tumors (3449 vs 145) demonstrates that replication-independent MMR, recently described at the molecular level (*Shell et al., 2007*; *Pena-Diaz et al., 2012*), is also involved in deamination repair in vivo (*Chen et al., 2014*). Finally, overall substitution frequencies correlated inversely with CpG islands. Indeed, irrespective of dinucleotide context, bases outside CpG islands were nearly two times more likely to undergo mutation than those inside CpG islands (*Figure 2H*). As CpG islands are generally unmethylated, DNA methylation thus appears to contribute to the mutagenic process. Explanations for this observation include the polymerase stalling that DNA methylation may induce (*Song et al., 2012*), and the repair of spontaneously deaminated methyl-Cs, which is error-prone and thus mutagenic on its own (*Chen et al., 2014*).

## Somatic indels in MMR-deficient hypermutators

We also evaluated somatic indel patterns in MMR-deficient tumors. As expected, since the majority of indels was located in homopolymers, a strong correlation between simple repeats and indel frequency was observed (*Figure 3A*). Indels were also predominantly 1 or 2 bps in length (*Figure 3B*). Although the minority of homopolymers consists of C or G bases (7%), an even smaller fraction of indels affected C:G homopolymers (1.9%; *Figure 3C*), suggesting that C:G homopolymers are less likely to accumulate indels. As observed in other MMR-deficient tumors and also in MMR-deficient *Caenorhabditis elegans* (*Denver et al., 2005*; *Kim et al., 2013*), deletions were remarkably more frequent than insertions (81% vs 19%), confirming that DNA polymerases are more prone to remove than to add a base during DNA synthesis.

## Exome-sequencing of additional MMR-deficient tumors

Next, we selected 13 additional MMR-deficient tumors, as well as four MMR-proficient tumors, collected from different tissues (i.e., endometrium, colon, and ovarium). Of these, six represented primary tumor cultures of low passage, which we preferred over cell lines, because the latter due to their hypermutator phenotype are no longer representative of the original tumor (*Figure 4—source data 2*).

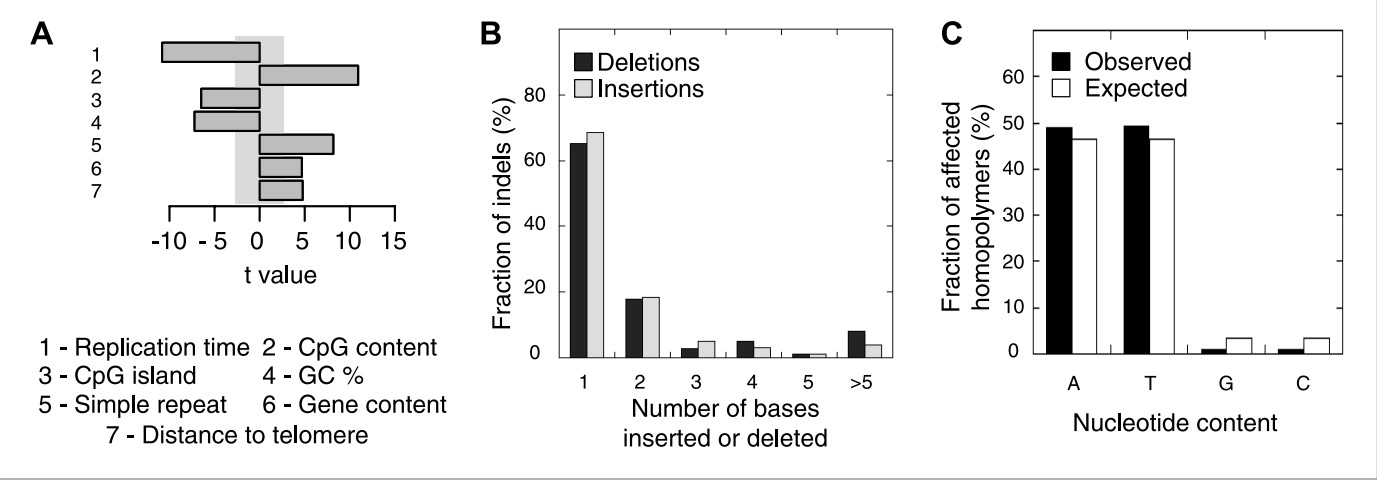

**Figure 3**. Somatic indel patterns in MMR-deficient tumors. (**A**) Impact of genomic features in MMR-deficient tumors on indel frequency as assessed by multivariate linear regression modeling. T-values resulting from the linear model are displayed for each genomic feature in the bar plots and indicate significance (shaded grey box equals p > 0.05, Bonferroni-corrected per model) and direction of the correlation. (**B**) Fraction of all indels inserting or deleting the indicated number of bases. (**C**) Fraction of homopolymers affected by an indel stratified per nucleotide, compared to the genome-wide fraction of homopolymers with that nucleotide content.

The following figure supplement is available for figure 3:

**Figure supplement 1**. The distance between a somatic substitution and the nearest somatic indel (top left), substitution (top right), repeat (bottom left), or homopolymer (bottom right) in the individual MMR-deficient genomes, and the expected distance based on 200 random models.

Exome-sequencing of tumor and matched germ-line DNA at an average coverage of 44x revealed that each MMR-deficient tumor contained ~2015 somatic events vs 39 for MMR-proficient tumors (52-fold increase; *Figure 4A*, *Figure 4—source data 1*, *Figure 4—source data 2*). Validation rates for substitutions and indels were respectively 87% and 86%. Clustering analysis of all 13 MMR-deficient tumors for the genes affected by either a somatic substitution or indel in the coding regions revealed no obvious subgroups in terms of cancer of origin or between primary tumors and cell cultures (*Figure 4—figure supplement 1*). Presumably, because of negative clonal selection and differences in homopolymer content in exons vs other genomic regions, exonic substitutions outnumbered indels (*Figure 4A*, *Figure 4—figure supplement 2*), similar to what we observed in the MMR-deficient whole-genomes (*Figure 1C,D*). Only a minority of these indels affected microsatellites, confirming that homopolymers were most frequently affected by indels.

Remarkably, 1.6% of homopolymers was recurrently affected by an indel in the 16 MMR-deficient tumors that underwent whole-genome or exome sequencing (i.e., 2244 out of 29,663 homopolymers were affected at least once, whereas 477 were affected at least twice; *Figure 4—figure supplement 3*). Furthermore, 34 and 10 homopolymers were affected in ≥6 or ≥8 tumors (*Figure 4—source data 3*). In contrast, only 55 substitutions were recurrent, three of which were found in ≥2 tumors (i.e., two substitutions affecting *KRAS* codon 12 and 13 were found in three and four tumors [*Tie et al., 2011*], whereas a substitution in *ZNF648* affected three tumors). When comparing homopolymer content of coding regions vs UTRs, long homopolymers (>10 bps) were more frequent in UTRs than in coding regions (*Figure 4B*). Because these long homopolymers were also more frequently affected (*Figure 4C*), the overall indel rate in coding regions was lower than in UTRs (*Figure 4D*). As a consequence of this difference, recurrent indels also occurred more frequently in UTRs than coding regions (31,438 vs 1337; *Figure 4—source data 3*). Remarkably, however, recurrent indels were more frequently observed than expected based on indel frequency in short, but not in long homopolymers (*Figure 4E*, *Figure 4—figure supplement 4*). This suggests that features other than homopolymer length underlie indel recurrence rates. Positive clonal selection of indels affecting short homopolymers, which are predominant in coding regions, represents a possible explanation. Very similar results were obtained when the analysis was repeated only on the 13 whole-exomes, indicating that exonic mutations identified from whole-genome sequences did not introduce any bias.

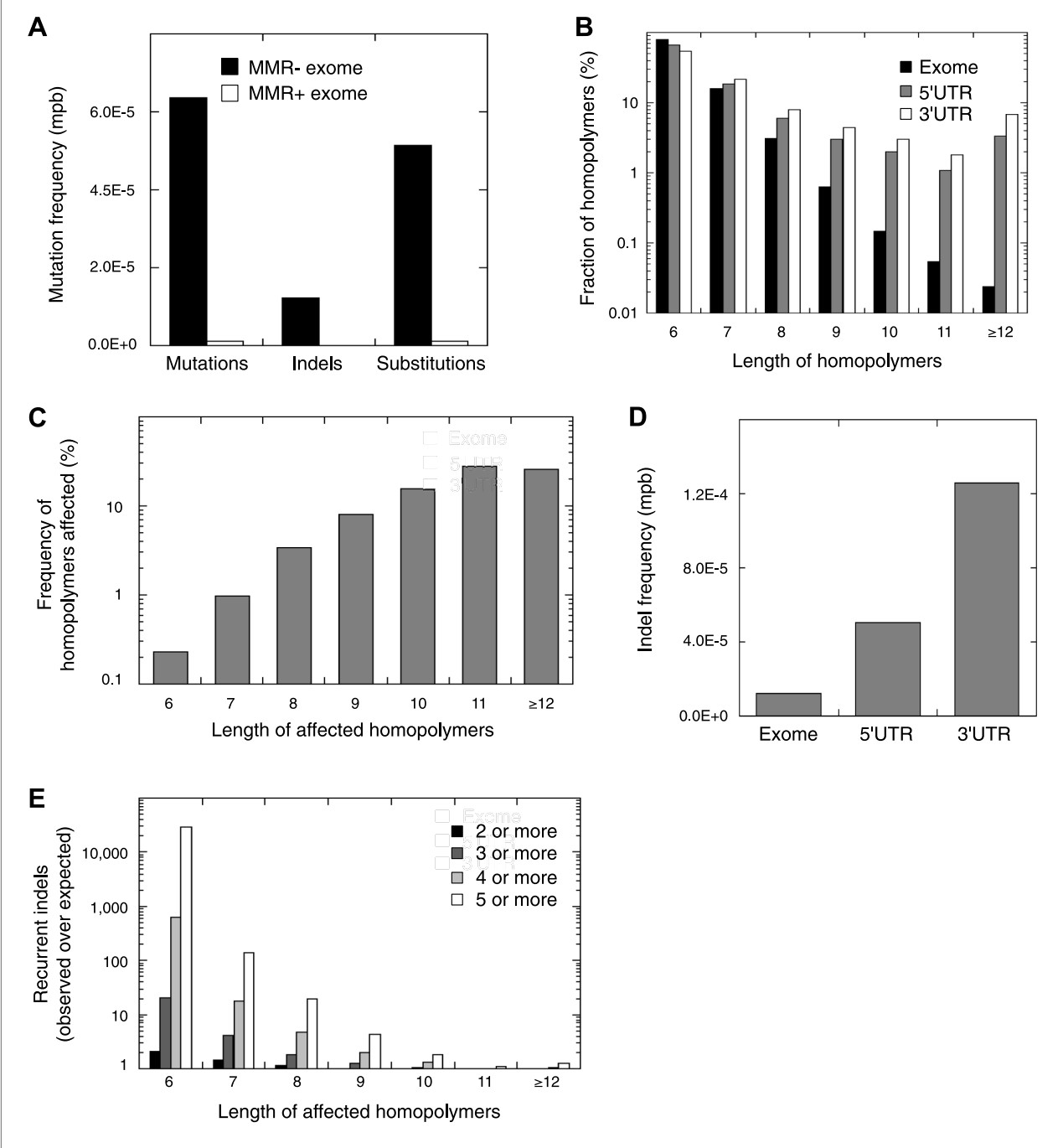

**Figure 4**. Recurrent somatic indels. (**A**) The average mutation frequencies in the exons of 13 MMR-deficient tumors and four MMR-proficient tumors. No obvious difference was observed between MLH1-, MSH2-, and MSH6- deficiency in terms of the mutation frequencies, substitution patterns, and indel compositions (***Figure 4—figure supplement 5***). (**B**) Fraction of homopolymers affected by an indel in function of the homopolymer length stratified for exons, 5′ and 3′UTRs. (**C**) The fraction of homopolymers in exons, 5′ and 3′UTRs that are affected by an indel in function of the homopolymer length. (**D**) Average somatic indel frequencies in exons, 5′ and 3′UTRs of 16 MMR-deficient tumors. (**E**) The enrichment of observed over expected frequencies of recurrent indels. Enrichments were stratified by length of the affected homopolymer and calculated for recurrent indels in 2, 3, 4, and 5 or more out of 16 MMR-deficient tumors.

The following source data and figure supplements are available for figure 4:

**Source data 1**. Sample info and sequence statistics of MMR-deficient whole exome sequenced tumour samples, a list of somatic substitutions detected therein and results of validation of somatic substitutions.

*Figure 4. Continued on next page*

*Figure 4. Continued*

**Source data 2**. A list of somatic indels detected in the MMR-deficient whole exome sequenced tumour samples, and results of their validation.

**Source data 3**. Somatic substitutions and indels in homopolymers together with their recurrence rate as identified by whole-exome and whole-genome sequencing.

**Figure supplement 1**. Clustering analysis of 13 MMR-deficient exomes for the genes affected by either a somatic substitution or indel in the coding regions.

**Figure supplement 2**. The fraction of indels (left panel) and substitutions (right panel) identified by whole-exome sequencing, as observed in microsatellites, homopolymers (length over 5 bp), short homopolymers (length of 3–5 bp) and 'not in repeat regions' compared to their expected fraction in these regions.

**Figure supplement 3**. Characteristics of the exonic homopolymers recurrently affected.

**Figure supplement 4**. The observed and expected frequencies of indels recurrently affected in homopolymers (in at least 2 out of 16 tumors) stratified for homopolymer length and for those affecting coding exonic regions and the 3'UTR.

**Figure supplement 5**. Mutation patterns obtained from MLH1-deficient, MSH2-deficient, and MSH6-deficient exomes.

## Recurrent indels reliably detect MSI in various cancer types

The extended Bethesda panel, which consists of eight microsatellite and two homopolymer markers, is currently used to diagnostically assess MSI (*Pinol et al., 2005*). This panel was historically compiled from a limited set of markers known to be variable. Due to their length and variability, these markers are notoriously difficult to analyze and interpret. As a consequence, the Bethesda panel has reduced sensitivity to detect MSI. In an effort to improve MSI testing, we randomly selected 59 recurrent indels affecting ≥6 out of 16 tumors; 50 markers were in 5' or 3'UTRs and 9 were in coding regions (*Figure 5—source data 1*). Furthermore, each of the markers was detected in both MMR-deficient EM and CRC. To facilitate high-throughput genotyping, the maximal length of affected homopolymers was restricted to 12 bps. First, we applied these 59 markers to a discovery set of 236 EM tumors for which MMR immunohistochemistry (IHC) data were available. This allowed us to determine three positive markers as the threshold with the best Matthew correlation coefficient to detect MMR-deficiency based on IHC and thus to define MSI (*Figure 5A,B*). At this threshold, our markers detected 40 out of 41 tumors MMR-deficient on IHC (sensitivity ~98%), while only 1 out of 184 MMR-normal tumors on IHC were identified as MSI (specificity > 99%). Notably, the latter patient had a familial history of cancer within the Lynch spectrum, suggesting that the tumor indeed exhibited MSI. Secondly, after having optimized the marker threshold, a head-to-head comparison against Bethesda panel was performed in 114 independent EM tumors as a validation. When observing discordances, we assessed MMR-deficiency using IHC to address which of both MSI panels was correct. Briefly, each MSI tumor on Bethesda (>2 markers positive) was also MSI with the 59-marker panel (*Figure 5C*). However, 12 tumors were positive in the 59-marker panel, but negative in Bethesda. IHC on the nine discordant tumors for which a paraffin block was available confirmed that each of them was MMR-deficient either for MLH1 or MSH2, indicating that the 59-marker panel has a higher sensitivity compared to Bethesda.

Likewise, we assessed MSI in 126 stage II or III CRC tumors. Each of the 28 MSI tumors on Bethesda was also positive with our 59-marker panel. In contrast, one tumor was MSI-positive in the 59-marker panel but not in the Bethesda panel (*Figure 5D*). This tumor contained a V600E BRAF mutation and was MLH1 hypermethylated, indicating that it was MMR-deficient and that our panel was also more sensitive for CRC (*Deng et al., 2004*). Finally, we also assessed whether our 59-marker panel can detect MSI in other cancer types. In a limited set of ovarian tumors and leukemias, we indeed correctly identified MSI in each of the samples tested (*Figure 5—source data 2*).

## MMR-deficient tumors are enriched in indels affecting DSB repair

Since we observed clear signs of clonal indel selection in MMR-deficient tumors, we assessed whether specific pathways were enriched for indels. We focused on frameshift indels in exons and exon/intron

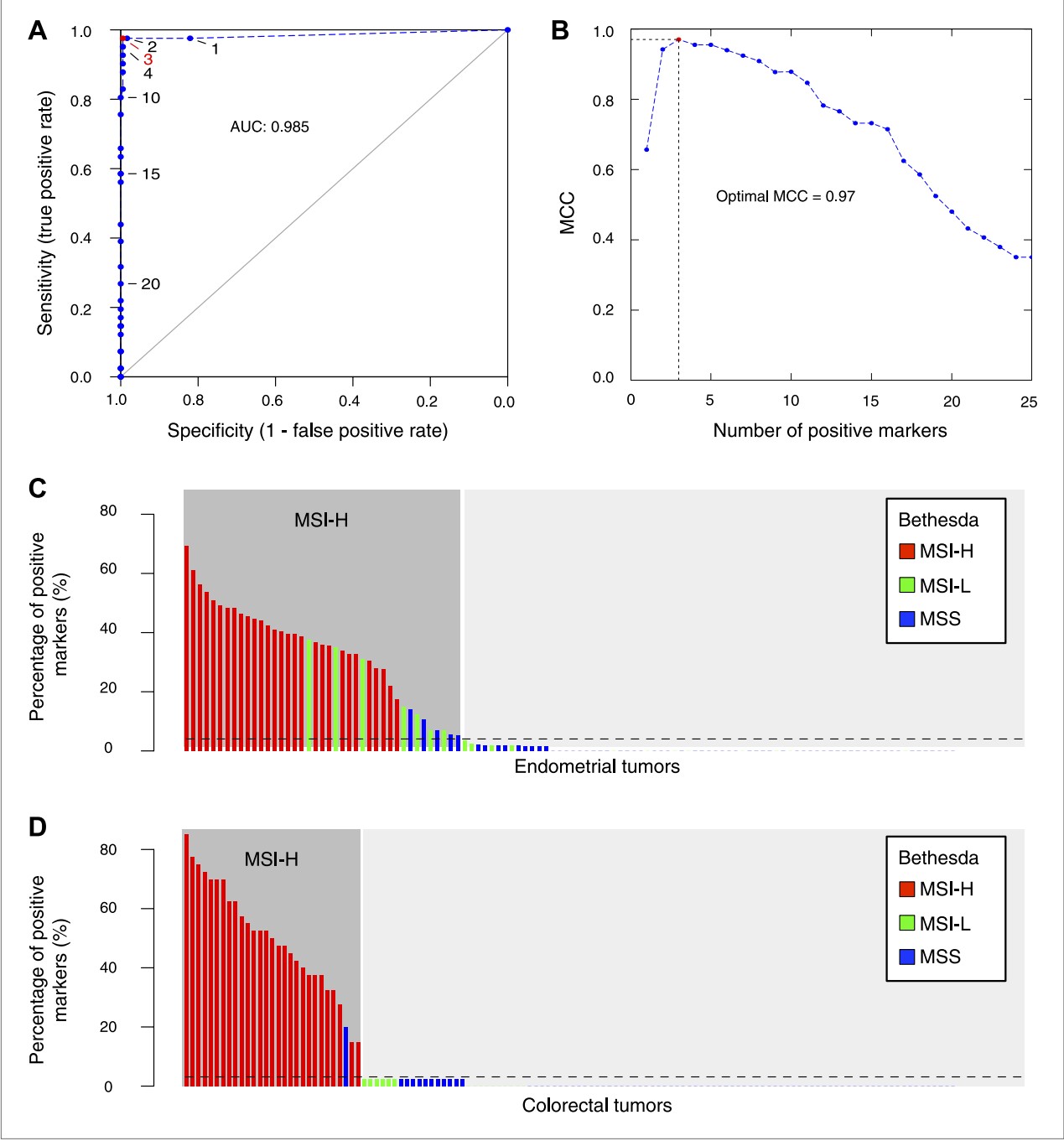

**Figure 5**. The 56-marker panel for MSI testing. (**A**) Receiver–operator curve assessing the impact of the number of positive homopolymer markers (out of 59) on the sensitivity and specificity of MSI testing, based on a panel of 236 EM tumors immunohistochemically characterized for their MMR status. (**B**) The Matthew Correlation Coefficient (MCC) of the ROC curve was calculated for each threshold, and a threshold of 3 resulted in the highest MCC-value (MCC = 0.97). (**C** and **D**) The extended Bethesda panel and the 59-marker panel were compared in an independent series of 114 unselected primary endometrial tumors (**C**) and 126 stage II or III CRC tumors (**D**). Results were color-coded according to high microsatellite instability (MSI-H; more than 1 markers positive), low microsatellite instability (MSI-L; 1 marker positive), or microsatellite stable status (MSS; 0 markers positive) as determined with the extended Bethesda panel. For endometrial tumors, 71 tumors (62%) were defined as MSS/MSI-L and 43 tumors (38%) as MSI-H by the 59-marker panel. Out of these 43 MSI-H tumors, Bethesda identified 32 tumors as MSI-H (>2 markers positive), 7 tumors as MSI-L, and 5 tumors as MSS. Vice versa, Bethesda did not identify any MSI-H tumor that was not identified by our panel. For colorectal tumors, there were 97 MSS tumors in our 59-marker panel that were concordantly called MSS or MSI-L by the Bethesda panel. The remaining 29 samples were detected as MSI in the 59-marker panel. 28 of these were also called MSI-H by the Bethesda panel, whereas one was called MSS by the Bethesda panel.

*Figure 5. Continued on next page*

*Figure 5. Continued*

The following source data are available for figure 5:

**Source data 1**. Recurrent indels selected for the 59-marker MSI panel and the results of a logistic regression analysis to detect differences between MSI-H and MSI-L/MSS tumors.

**Source data 2**. Clinical information, MMR-mutation status and sequencing statistics for ovarian tumors and leukemias.

boundaries as they represent loss-of-function mutations (*Ham et al., 2006*), and thus have a less ambiguous functional impact than indels in UTRs. On average, each MMR-deficient tumor contained 472 such indels, 59 of which were recurrent indels. Pathway analyses using IPA of all genes affected by a somatic indel, excluding the core MMR genes, ranked the '*Role of BRCA1 in DNA damage response*' as the top enriched pathway. IPA analysis of genes affected by recurrent indels moreover revealed that the '*Double-strand break repair by homologous recombination*' pathway (*DSBR by HR*) ranked top (*Table 2*). We also performed pathway analyses using the more advanced GenomeMuSiC, which takes background mutation rates into account and assigns weights depending on the number of tumors and genes affected in a given pathway. GenomeMuSiC analyses based on either the independently assembled Reactome or BioCarta pathway databases, ranked respectively the '*ATR/BRCA pathway*' and the *DNA repair*' pathway first, with the more specific '*Homologous recombination repair*' pathway ranking third in the latter (*Table 2*). Based on an expert curated DNA repair database (DNARepairDB), '*Homologous recombination*' represented the only DNA repair pathway that was significantly enriched in indels. Since each pathway database differed with respect to the genes included, we finally compiled a literature-based set of genes with proven involvement in *DSBR by HR*, allowing us to more accurately estimate that each MMR-deficient tumor on average contained 3.3 ± 0.4 indels in the '*DSBR by HR*' pathway (*Table 2*, *Table 2—source data 1*). Notably, none of the top-ranking pathways for any of the databases contained significantly more homopolymers in their genes than expected.

In an effort to replicate these findings, we analyzed mutation data of 27 CRC and 65 EM tumors with MSI sequenced by The Cancer Genome Atlas (*TCGA, 2012*; *Kandoth et al., 2013*). Although most of these tumors were sequenced at low coverage depth, we identified 2183 and 3138 mutated genes from respectively the CRC and EM tumor data sets. IPA analysis confirmed that the *Role of BRCA1 in DNA damage response* was again amongst the top enriched pathways for each of the data sets. The corresponding p-values were 9.06E−3 and 2.97E−4, although only the latter survived multiple testing correction (p = 0.022; *Table 2—source data 1*). As raw data sets were not accessible, the more sensitive GenomeMuSiC could not be used.

## Reduced DSBR by HR activity in primary MMR-deficient cells

Homozygous mutations affecting genes in the *DSBR by HR* pathway cause DSB repair defects reminiscent of BRCA1 or BRCA2 loss, a phenotypic feature dubbed *BRCAness* (*McCabe et al., 2006*). Having established that MMR-deficient tumors are enriched in heterozygous frameshift mutations in the *DSBR by HR* pathway, we investigated the functional impact of these events. First, we confirmed that indels affecting the *DSBR by HR* pathway were located in the major tumor subclone (*Table 2*, *Table 2—source data 1*). Then, we analyzed HR in seven MMR-deficient and four MMR-proficient patient-derived primary tumor cultures. We exposed these cultures to the PARP inhibitor olaparib, which induces DSBs upon DNA replication through single-strand break repair inhibition, and to mitomycin C, which induces DSBs through DNA cross-links and replication fork collapse (*Bunting et al., 2012*). We then quantified the relative number of cells with γH2AX- and RAD51-positive foci, respectively, as a measure of induced DSBs and ongoing HR. Exposure to olaparib or mitomycin C triggered an increase in γH2AX-foci in all tumor cultures, regardless of MMR status. In contrast, although RAD51 foci formation was evident in MMR-deficient and MMR-proficient cultures, the increase was far less pronounced in MMR-deficient cultures (*Figure 6A,B*), and this for both olaparib (p = 0.021) and mitomycin C (p = 0.006) exposure. The reduction in RAD51 foci could not be ascribed to differences in RAD51 protein expression or differences in cell cycle between MMR-deficient and -proficient cells, as these were similar between both sets of cultures, under both treated and untreated conditions (*Figure 6—figure supplements 1–3*). Since RAD51 foci are completely absent upon PARP inhibition in cells with homozygous loss of *BRCA1*, but not affected in heterozygous mutation carriers (*Farmer et al., 2005*), these ex vivo data suggest that the

**Table 2.** Pathways most significantly affected by exonic indels

| Database | Pathway | Rank | FDR | Affected samples (n = 16) | Mutations per sample |
|---|---|---|---|---|---|
| | *DSBR by HR (custom definition) | n.a. | n.a. | 16 | 3.25 |
| BioCarta (ranking by GenomeMusic) | *ATR/BRCA pathway | 1 | 1.0E−16 | 15 | 3.50 |
| | ATM pathway | 2 | 5.9E−11 | 15 | 2.69 |
| | G2 pathway | 3 | 7.2E−08 | 15 | 2.81 |
| | IL10 pathway | 4 | 2.2E−05 | 12 | 1.75 |
| | CARM1 and regulation of the Estrogen Receptor pathway | 5 | 2.2E−05 | 14 | 3.19 |
| DNA Repair DB (ranking by GenomeMusic) | *Homologous recombination pathway | 1 | 1.3E−04 | 13 | 1.56 |
| | Base excision repair pathway | 2 | 9.0E−02 | 10 | 0.75 |
| | Non-homologous end joining pathway | 3 | 1.7E−01 | 9 | 0.69 |
| | Nucleotide excision repair pathway | 4 | 8.3E−01 | 7 | 0.50 |
| Reactome (ranking by GenomeMusic) | DNA repair | 1 | 2.5E−11 | 15 | 6.69 |
| | Double strand break repair | 2 | 7.2E−08 | 15 | 2.94 |
| | *Homologous recombination repair | 3 | 1.9E−07 | 15 | 2.31 |
| | G2/M checkpoints | 4 | 2.3E−07 | 15 | 3.50 |
| | Cell cycle checkpoints | 5 | 4.5E−05 | 15 | 4.75 |
| | Base excision repair | 15 | 8.3E−03 | 10 | 0.94 |
| | Non-homologous end joining | 59 | 1.0E+00 | 8 | 0.63 |
| | Nucleotide excision repair | 61 | 5.9E−01 | 10 | 1.50 |
| IPA (ranking by IPA) | *DNA double-strand break repair by homologous recombination | 1 | 4.7E−03 | 15 | 1.56 |
| | Ovarian cancer signaling | | 4.7E−03 | 16 | 5.75 |
| | Role of NFAT in cardiac hypertrophy | 3 | 6.8E−03 | 14 | 3.88 |
| | Cell cycle: G2/M DNA damage checkpoint regulation | 4 | 1.3E−02 | 15 | 2.88 |
| | PPARα/RXRα activation | 5 | 1.4E−02 | 15 | 4.63 |
| | DNA double-strand break repair by non-homologous end joining | 60 | 1.7E−01 | 14 | 1.50 |

The five top ranking pathways are listed, as well as all annotated pathways relevant for DNA repair. The custom definition used throughout this manuscript was added for illustrative purposes. n.a. = not applicable.
*The DSBR by HR pathway.

**Source data 1**. Results of pathway enrichments, custom definition of the DSBR by HR pathway and the allelic frequencies of mutations in HR genes.

accumulation of indels in MMR-deficient tumors gradually impairs the *DSBR by HR* pathway to a level that is intermediate to that of cells heterozygous- and homozygous-deficient for BRCA1.

## DSB inducers sensitize MMR-deficient tumors

As MMR-deficient tumors are compromised in their *DSBR by HR* activity, we wondered whether these tumors, similar to BRCA1-deficient tumors (*Farmer et al., 2005*), are more sensitive to agents that induce DSBs. First, since PARP inhibitors are already used in clinical practice, all seven MMR-deficient and four MMR-proficient cultures were dose-dependently exposed to olaparib. This revealed that MMR-deficient cultures exhibited a dose-dependent decrease in proliferation upon exposure to olaparib, whereas MMR-proficient cultures were only affected at higher concentrations. Likewise, cell cytotoxicity assays revealed a dose-dependent sensitivity of MMR-deficient cells to olaparib that was more pronounced than in MMR-proficient cells (50% growth inhibition [GI50]) was reached at 26 μM vs 129 μM,

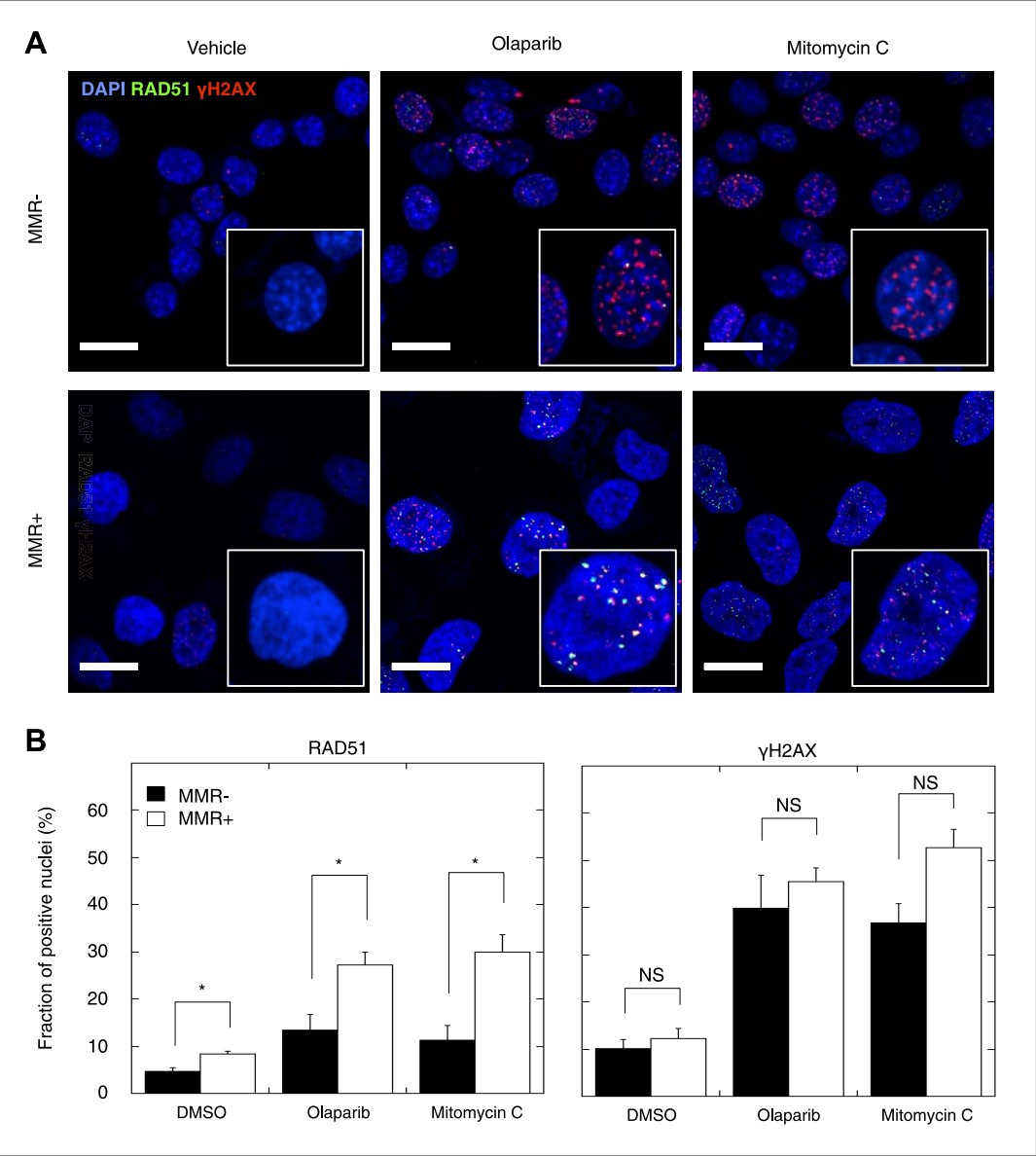

**Figure 6**. Reduced DSBR by HR activity in MMR-deficient cells. (**A**) Representative confocal images of MMR-deficient and MMR-proficient primary tumor cells exposed for 24 hr to vehicle, 26 µM olaparib, or 300 nM mitomycin C stained for the homologous repair marker RAD51 (green), the DNA damage marker γH2AX (red), and counterstained with DAPI (blue). The bar is 10 µm wide. (**B**) Quantification of cells containing >5 RAD51 or γH2AX foci. Averages are shown for MMR-deficient and MMR-proficient primary tumor cultures after 24 hr of treatment with vehicle, 26 µM olaparib or 300 nM mitomycin C.

The following figure supplements are available for figure 6:

**Figure supplement 1**. Cell cycle distribution in untreated MMR-deficient and MMR-proficient cell cultures.

**Figure supplement 2**. MMR-deficient tumor cultures were challenged with olaparib (26 µM), camptothecin (30 nM), or mitomycin C (300 nM) for 24 hr, pulsed with BrdU for 2 hr and analyzed for cell cycle by propidium iodide staining (DNA content analysis) using flow cytometry.

**Figure supplement 3**. Example of a 2 hr BrdU pulse-labeled MMR-deficient cell culture, demonstrating S-phase stalling and G2/M stalling upon mitomycin C exposure, S-phase stalling upon camptothecin exposure and S-phase stalling and G2/M stalling upon olaparib exposure.

respectively, p = 0.0064 (*Figure 7A,B*, *Figure 7—figure supplement 1*). Other DSB-inducing compounds such as mitomycin C or ionizing radiation similarly proved more detrimental for MMR-deficient than MMR-proficient cells (*Figure 7B*). In contrast, cytotoxicities of other chemotherapeutic compounds such as paclitaxel were comparable between both groups.

Finally, in order to more accurately measure the level of HR-deficiency in MMR-deficient tumors, we assessed the level of knock-down of BRCA1, BRCA2, and ATR needed to achieve an olaparib sensitivity similar to that observed in MMR-deficient cells, that is, a GI50 of 26 µM. BRCA1, BRCA2, or ATR expression was dose-dependently reduced using siRNAs in the MMR- and HR-proficient cell line, MCF7. A growth inhibition of 50% was reached in MCF7 cells when applying 5.9 nM ATR, 0.88 nM BRCA1 or 0.41 nM BRCA2 siRNA, corresponding respectively to a reduction in expression of 69.5 ± 1.1%, 76.1 ± 4.4%,

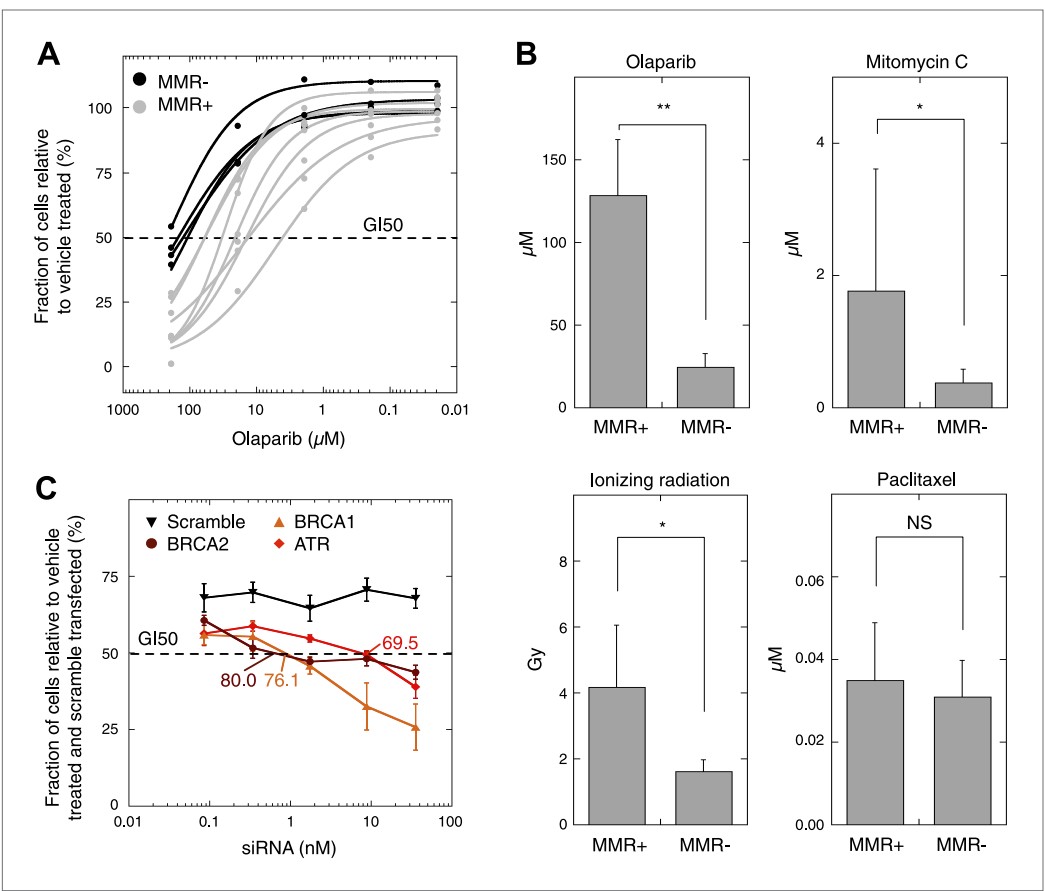

**Figure 7**. MMR-deficient cells are sensitive to PARP inhibition. (**A**) Dosimetry experiments assessing the effect of decreasing concentrations of olaparib on in vitro cell proliferation relative to the corresponding untreated cultures as measured by sulforhodamine B assays. (**B**) Cytotoxicity of olaparib, mitomycin C, ionizing radiation and paclitaxel as measured by sulforhodamine B assays. Displayed are the average concentrations (µM) or dose (Grey, Gy) that inhibit 50% of the normal growth. p-values are 0.0077, 0.040, and 0.038 for olaparib, mitomycin C, and ionizing radiation, while p-value is not significant (NS) for paclitaxel. (**C**) Effect of knock-down of BRCA1, BRCA2, and ATR mRNA on olaparib sensitivity of the MMR-proficient, HR-proficient MCF7 cell line. Cells were transfected with the indicated siRNA concentration (X axis), and after 24 hr incubated with 26 µM olaparib or vehicle. Another 48 hr later, cell viability was assessed using the sulforhodamine B assay. The siRNA concentration corresponding to a growth inhibition of 50% was subsequently assessed for the level of knock-down induced. The resulting values are indicated on the plots and are expressed as %. Values plotted were normalized to vehicle-treated cells transfected with a scrambled siRNA of matching concentration.

The following figure supplement is available for figure 7:

**Figure supplement 1**. Cell proliferation of MMR-deficient cultures was measured in real-time using the xCELLigence RTCA DP system (for up to 48 hr after treatment).

and 80.0 ± 2.4% (*Figure 7C*). These data thus suggest that the loss of *DSBR by HR* activity in MMR-deficient tumors corresponds to a loss of about 75–80% BRCA1 or BRCA2 expression.

## Discussion

Here, we surveyed whole-genomes of MMR-deficient tumors to provide a comprehensive picture of the mutations associated with human MMR-deficiency. With respect to somatic substitutions, we observed that the majority represented transitions and not transversions, and that adjacent nucleotides and various genomic features had an important context-dependent effect on determining which nucleotides were affected. Remarkably, the observed substitution pattern, in particular how it was impacted by small and large-scale contexts, was very similar to that in the germ-line at different time scales: for germ-line substitutions as they currently arise (de novo), as they have accumulated in the human population or as they served as a substrate for human-chimpanzee divergence (*Hodgkinson and Eyre-Walker, 2011*). Our observations thus suggest that, similar to bacterial populations and other lower organisms (*Saint-Ruf and Matic, 2006*), incomplete mismatch repair in humans contributes significantly to genetic variability and probably also to natural selection through genetic adaptation. Additionally, our data provide fundamental insights into the function of the MMR machinery. We observed, for instance, a higher number of substitutions in methylated CpG sequences, implicating MMR in the repair of methylated cytosine deamination and demonstrating that MMR disconnected from the replication fork is also critical to maintain genomic integrity.

At the whole-genome level, ~80% of somatic mutations represented indels. Although indel detection using high-throughput sequencing is burdened with high false-positive rates, 88.0% of the indels identified here validated favorably using orthogonal technologies. When focusing on the clinical relevance of indel mutation patterns to diagnose MSI, we observed that indels specifically affected homopolymer stretches, which is relevant as the extended Bethesda panel consists of eight microsatellite and only two homopolymer markers and possibly therefore has only limited sensitivity relative to IHC (~75% for both EM and CRC tumors [*Hampel et al., 2005*, *2006*, *2008*]). Our 59-marker panel consisting only of markers in homopolymers was clearly more sensitive than Bethesda, yielding sensitivity rates of 87% relative to IHC. This was not due to the fact that we genotyped more markers than Bethesda, as restricting our panel to 10 markers still resulted in a sensitivity rate of 85% (data not shown). Furthermore, since our panel was based on recurrent mutations present in both CRC and EM, and since 50 out of 59 markers were located in UTRs, which are less likely to drive clonal selection and thus to represent tissue-specific events, it could be used to detect MSI in cancers affecting various tissues. Finally, since all markers were located in homopolymers ≤12 bps in length, they are, in contrast to the 25 or 26 bps markers from Bethesda, compatible with various low- to high-throughput genotyping technologies, thereby greatly facilitating their clinical adoption. For instance, we were able to multiplex all 59 markers in just five PCR amplification reactions compatible with Sequenom MassArray genotyping.

Pathway analyses on all genes affected by exonic indels further revealed that the *DSBR by HR* pathway was enriched for somatic indels. Although mutations in genes involved in this pathway, such as *MRE11A* or *RAD50*, have previously been reported in MMR-deficient tumors, these studies focused on specific mutations in individual genes rather than on pathways, and for this reason could establish that only a fraction of MMR-deficient tumors was affected by mutations in these genes (*Miquel et al., 2007*). In contrast, our study identified that every MMR-deficient tumors was affected by on average 3.3 somatic indels in the *DSBR by HR* pathway. Furthermore, although it is well established that cells deficient in BRCA1, BRCA2, Fanconi anemia, or other HR-related genes are hypersensitive to DSB inducers (*Murai et al., 2012*), as for instance, synthetic lethality in BRCA1- or BRCA2-deficient tumors through PARP inhibition is already approved as therapy in breast and ovarian cancer (*Metzger-Filho et al., 2012*), data demonstrating sensitivity of MMR-deficient cells to DSB inducers have not been conclusive (*Takahashi et al., 2011*; *Vilar et al., 2011*; *Park et al., 2013*). For instance, although there are some reports highlighting the sensitivity of MSH3-deficient cell lines to DSB inducers, this appeared to occur through a non-canonical MMR pathway, as MLH1 was not involved in this process (*Takahashi et al., 2011*; *Park et al., 2013*). Furthermore, the only clinical study set-up so far to explore efficacy of PARP inhibitors as a single-agent therapy in previously treated patients with metastatic CRC stratified by MSI status, was unfortunately delayed due to patient accrual issues.

Our hypothesis-free discovery that *DSBR by HR* is the top pathway affected by heterozygous loss-of-function mutations in MMR-deficient tumors, both in our own data set and TCGA, also suggests that mutations in *DSBR by HR* genes converge in an oligogenic model, wherein the number of indels

dose-dependently decreases *DSBR by HR* activity, thereby rendering them gradually more sensitive to DSB inducers. As a result of this double-hit, our ex vivo culture experiments are, however, difficult to compare to experiments relying on genotype-matched cells that have a single hit in the MMR or HR pathway. In addition, since *MMR* and *DSB by HR* pathway activities are not characterized in a clinical setting, it is difficult to relate our data to clinical studies assessing the outcome of therapeutics such as cisplatin or 5-fluorouracil, which have potential opposing activities on MMR- and HR-deficient tumors.

Clinical studies are therefore needed to assess whether DSB inducers, such as PARP inhibitors, are indeed also effective in MSI tumors. In particular, since on average 3.3 heterozygous loss-of-function mutations only partially inactivate the DSB repair by HR pathway (~80% inactivation), it remains to be seen whether, compared to BRCA1 or BRCA2-deficient tumors, in which the HR pathway is completely inactivated, clinically relevant benefits are also achievable in MSI tumors. Possibly, only those MMR-deficient tumors containing large numbers of indels (≥5) in the *DSBR by HR* pathway will show a significant response. Nevertheless, there is a great clinical need for novel treatment options in MSI tumors. Indeed, although stage II or III CRC tumors with MSI are characterized by a modestly improved prognosis, MSI tumors in the advanced setting are generally associated with a more peritoneal metastasis and a worse overall survival independent of the chemotherapy regimen (*Smith et al., 2013*; *Yoon et al., 2013*). Our observations thus clearly warrant novel clinical studies assessing the therapeutic efficacy of DSB inducers in MMR-deficient tumors.

## Materials and methods

### Standard diagnostic tests for MMR-deficiency

To assess MLH1-, MSH2-, and MSH6-deficiency immunohistochemistry using monoclonal antibodies against MLH1 (clone ES05; DAKO, Heverlee, Belgium), MSH2 (clone G219-1129; BD Pharmagen, Erembodegem, Belgium), and MSH6 (clone EP49; Epitomics, Burlingame, USA) were applied. Absence of nuclear staining in tumor cells and normal staining in the surrounding normal tissue were considered as MMR-deficient. Methylation of the *MLH1* promoter was determined using the SALSA MS-MLPA KIT (MRC-Holland, Amsterdam, The Netherlands). PCR reaction fragments covering the Deng C and Deng D regions were separated by capillary gel electrophoresis (ABI 3130; Applied Biosystems, Ghent, Belgium) and quantified using the Genemarker (v1.91) software (Softgenetics). MSI status was detected by the extended Bethesda panel using capillary gel electrophoresis, as described previously (*Dietmaier et al., 1997*; *Boland et al., 1998*).

### Sample selection and preparation

We selected 17 endometrial, three colorectal, and two ovarian tumor–normal pairs for either whole-genome or whole-exome sequencing. Samples were all chemo-naive. DNA was derived from fresh frozen, primary tumors. Matched normal DNA for these 22 samples was extracted from peripheral white blood cells.

### Whole-genome sequencing, analysis, and annotation

Five tumor–normal pairs were selected for whole-genome sequencing. Paired-end sequencing was performed using the Complete Genomics service (CG, Mountain View, California, USA) as described in *Drmanac et al. (2010)* or by Illumina HiSeq2000. For CG sequencing, reads were initially mapped to the reference genome (hg18) using Complete Genomics' CGAtools. Between 207 and 338 Gb of sequencing data were obtained, resulting in a haploid coverage between 73× and 119×. Approximately, $2.7 \times 10^9$ bases were called in each genome, representing ~95% of the total genome and ~97% of the exome. Substitutions and indels were called by the variant caller in the CGAtools. On average, 3,132,715 substitutions and 357,153 indels were detected in each genome. The CGAtool (v1.0.3.9) *calldiff* method was used to detect somatic mutations in the tumor–normal pairs. For Illumina sequencing, 2 × 100 bp paired-end sequencing was performed, yielding 25–30x coverage per sample. Burrows-Wheeler Alignment (BWA) was used to align the raw reads to the reference genome (hg19) (*Li and Durbin, 2010*). PCR duplicates were removed with Picard MarkDuplicates (v1.32). Base recalibration, local realignment around indels and single nucleotide variant calling were performed using the GenomeAnalysisToolKit (GATK v1.0.4487) (*McKenna et al., 2010*). Small indels were detected using Dindel (v1.01) (*Albers et al., 2011*). Substitutions and indels with quality score >Q30 were considered. On average, 3,977,086 substitutions and 837,915 indels were detected in each genome. Somatic mutations were detected by means of *intersectBed* command of BEDTools (v2.12.0) (*Quinlan and*

*Hall, 2010*). Raw data for all whole-genomes are available under restricted access in the European Genome-Phenome Archive (EGA) with accession number EGAS00001000182.

Sequence data were annotated using ANNOVAR (v2013Jun21) and the UCSC RefGene annotation track. Germ-line substitutions and indels were eliminated from the list of somatic mutations using the following publicly available datasets: (i) common SNPs in dbSNP (v132) with a minor allele frequency of >1%, (ii) substitutions identified in the November 2010 release of the 1000 Genomes Project, (iii) the Axiom Genotype Data Set containing common SNPs from 1261 HapMap3 individuals in 11 populations, and (iv) variant data identified in 46 HapMap individuals (CG diversity panel). Somatic mutations were validated using Sequenom MassARRAY genotyping, as previously described (*Reumers et al., 2011*). Details of validation experiments are shown in *Figure 1—source data 3*. A quality score method to enrich for true somatic mutations by defining a threshold that differentiates false-positive and true-positive variants based on Sequenom validation data was applied to CG genomes and increased the validation rate for substitutions from 93.5%, 71.4%, and 55.6% to 97.7%, 100%, and 73.3% for MMR− 1, MMR+ 1, and MMR+, 2 respectively. Detailed data of all somatic mutations are in *Figure 1—source data 1* and *Figure 1—source data 2*. Copy number status of the sequenced tumors was determined by Illumina CytoSNP-12 chips and analyzed using the ASCAT algorithm (*Van Loo et al., 2010*). Copy number status of the five whole-genomes was shown as *Figure 1—figure supplement 3*.

## Genome annotation

The genome was annotated into the following functional genomic regions: (coding) exonic regions (1.12%), intronic regions (34.01%), 3′ untranslated regions (3′UTR, 0.78%), 5′ untranslated regions (5′UTR, 0.14%), noncoding RNA (ncRNA, 2.81%), upstream genic regions (defined as 1 kb before the start of the gene, 0.58%), downstream genic regions (defined as 1 kb after the end of the gene, 0.58%), and intergenic regions (59.98%).

## Evidence of negative clonal selection

Overall mutation frequencies were defined as the number of somatic mutations per base (mpb) in a given genomic region. To assess negative selection in the exome, we checked whether (i) there was a lower mutation frequency in the exome relative to the whole-genome, and whether (ii) the frequency of somatic mutations was more prominently decreased in the exome. As homopolymers in exomes have characteristics that differ from those in the rest of the genome in terms of number, base composition and length, we corrected indel frequencies for these confounding factors. We calculated the frequency of affected homopolymers for each genomic location (**t**: exonic, 5′UTR, 3′UTR, intronic, intergenic, or genomic), for each type of homopolymer (**AT** or CG composition) and each homopolymer length (6, 7, 8, etc[**l**]). $^{AT}Freq^t_l = {}^{AT}_{aff}n^t_l$. Next, we calculate the relative increase of observed frequencies relative to the frequency observed at the genome-wide level: $^{AT}rFreq^t_l = {}^{AT}Freq^t_l/{}^{AT}Freq^{genome}_l$. The frequency $^{AT}rFreq^t_l$ was normalized for the number of homopolymers of a given length l, for each genomic location t and for homopolymer composition ($^{AT}wrFreq^t_l = {}^{AT}rFreq^t_l \times {}^{AT}n^t_l/\sum {}^{AT}n^t_l$), and further normalized for the number of AT (or GC) homopolymers for each genomic location and homopolymer length ($^{AT}nwrFreq^t_l = {}^{AT}wrFreq^t_l \times {}^{AT}n^t_l/({}^{AT}n^t_l + {}^{CG}n^t_l)$). All the weighted frequencies are then summed for every genomic location ($cFreq^t = \sum {}^{AT}nwrFreq^t_l + \sum {}^{CG}nwrFreq^t_l$) and divided by the overall summed genomic frequency ($rFreq = cFreq^t/cFreq^{genomic}$).

## Data sets of germ-line and somatic variants

The following datasets were used: (i) the 1000 Genomes Project containing common variants with a minor allelic frequency >10%, (ii) all germ-line variants identified in the 3 MMR-deficient tumors sequenced in this study, (iii) de novo mutations from 83 trios as published by *Campbell et al. (2012)* and *Kong et al. (2012)*, and (iv) a human-chimp divergence set of substitutions as previously described (*Stamatoyannopoulos et al., 2009*). Somatic mutations identified in other tumor whole-genomes were: (i) BRCA-deficient breast cancer tumors as published by *Nik-Zainal et al. (2012)*, (ii) MMR-proficient endometrial tumors sequenced in this study, (iii) melanoma genomes as published by *Pleasance et al. (2010)*, and (iv) small cell lung cancer (SCLC) as published by *Pleasance et al. (2010)*.

## Genomic features postulated to underlie the systematic variation of mutation rates

The distance to telomere was defined as the distance from the middle of the 1 Mb window to the beginning or the end of the chromosome whichever was the shortest. Replication time was considered as published by *Chen et al. (2010)*. Simple repeats represented the number of homopolymer and

microsatellite bases. GC% was calculated as (G+C)/(A+T+G+C), CpG content as the number of CG dinucleotide bases, CpG islands as the number of bases belonging to CpG islands, gene content as the number of bases belonging to each genomic region. DNase hypersensitivity (DNaseI size) and nuclear lamina binding sites were downloaded from UCSC and the number of bases per site was counted for both.

## Exome-sequencing, analysis, and annotation

We sequenced 11 tumor–normal pairs, 6 primary cell cultures (PC) and their match normal DNA samples. Detailed clinical information is shown in *Figure 4—source data 1*. Exomes were captured using Illumina's TruSeq Exome Enrichment Kit. The TruSeq capture regions encompass 62 Mb, consisting of 94.4%, 83.9%, and 91.9% of the exonic, 5'UTR and 3'UTR regions respectively. 2 × 75 bp paired-end sequencing reactions were used for all EM tumors, while 2 × 100 bp paired-end sequencing was performed on CRC tumors and PC samples. Analysis, annotation, and validation were performed similarly as for whole-genome sequencing. On average, the coverage was 44.5× and 95.1% of bases were called in the captured regions, yielding 51,782 substitutions and 30,290 indels per sample. Raw data are available under restricted access in EGA under accession number EGAS00001000182. Details of validated somatic mutations are available in *Figure 4—source data 1* and *Figure 4—source data 2*.

## Recurrent somatic mutations

The 13 MMR-deficient whole-exomes and whole-exome data extracted from 3 MMR-deficient whole-genomes were screened for recurrent mutations. Random selection and validation of 24 indels occurring in 6 or more samples revealed a validation rate of 100%. Given the high validation rate for somatic indels per se, and the even higher rate for recurrent indels, we considered all recurrent indels as true-positives. Subsequent analyses were limited to indels recurrently affecting homopolymer regions, that is, 29,663 Illumina TruSeq-captured exonic homopolymers. Details of recurrent mutations in these homopolymers are available in *Figure 4—source data 3*. We also screened 5430 and 60,942 homopolymers located in the exome-captured 5' and 3' UTRs for recurrent indels. Details of these recurrent indels are in *Figure 4—source data 3*. Recurrent indels meeting the following criteria were considered for a targeted Sequenom panel assessing MSI: (i) occurring in 6 or more samples, (ii) detected in both EM and CRC exomes, (iii) the maximal length of affected homopolymer <12 bp. After extensive optimization experiments, 59 markers were chosen. Detailed information about each indel is given in *Figure 5—source data 1*.

## MSI panel

236 EM tumors used to establish MSI thresholds were drawn from the Australian National Endometrial Cancer Study (ANECS). IHC analyses of these tumors were independently performed at the Molecular Cancer Epidemiology Laboratory in Brisbane, Australia as described (*Tan et al., 2013*). 11 out of 236 tumors were excluded for the 59-marker panel due to their low tumor percentage (≤10%). By varying the marker threshold, we calculated the number of true-positives and false-positives identified by our MSI panel relative to the IHC data. A ROC curve was constructed based on these values. The Matthew Correlation Coefficient of the ROC curve was calculated for each threshold. Tumors were considered MSI when they had three markers positive. We did not distinguish between MSI-low and microsatellite stable (MSS), as this is currently not clinically relevant. All tumors with less than three positive markers were thus considered MSS/MSI-L. For the Bethesda panel, we defined three categories as follows: microsatellite stable (MSS, 0 out of 10 markers), low microsatellite instability (MSI-L, 1–2 out of 10 markers), and high microsatellite instability (MSI-H, 3 or more out of 10 markers). Two sets of data (114 EM tumors and 97 CRC tumors) were used for the comparison. Details of these sample sets are given in *Figure 5—source data 1*.

## Mutation signatures in other tumor types

The 59-marker panel was applied to ovarian tumors and leukemia. Four samples with proven MSI status were selected, including one ovarian tumor (OV) and three leukemia cell lines (DND41, CCRF-CEM, and SUPT1). The MSI-H OV tumor, two MSS OV tumors, and their matched normal samples, as well as three MSI-H leukemia cell lines and a MSS leukemia cell line (RPMI-8402) were exome-sequenced. Detailed information for all samples can be found in *Figure 5—source data 2*. Raw data are available in EGA under the accession number EGAS00001000182.

## Pathway analyses for recurrent mutations

Two pathway tools (IPA and GenomeMuSiC) and three pathway databases (IPA, BioCarta, and Reactome [*Haw et al., 2011*]) were used. We first selected all genes with somatic exonic indels, and then extended our mutation calling to indels occurring 25 bp up or down-stream of each exon. Mutation calling and filtering for the later set of mutations was done as described above. In total, 1989 additional indels in exon/intron boundaries were detected. These were combined with the previously described indels in exonic regions, which—after the removal of indels in MMR genes—yielded 7546 indels in 4116 genes. As a validation, we selected 27 CRC and 65 EM tumors with MSI sequenced by The Cancer Genome Atlas (*TCGA, 2012*; *Kandoth et al., 2013*). We selected genes recurrently affected not only by frameshift indels but also by non-synonymous substitutions. There were 2183 and 3138 genes from the CRC and EM tumor data sets, respectively. Detailed results of pathway analyses are given in *Table 2—source data 1*.

## Establishment of primary tumor cell cultures

11 primary endometrial and ovarian tumor cell cultures were established from tumors of patients undergoing surgery at the Division of Gynecologic Oncology, UZ Leuven (Belgium). Tissue was washed with PBS supplemented with penicillin/streptomycin and fungizone, digested with collagenases type IV (1 mg/ml; Roche, Vilvoorde, Belgium) and DNAse I (0.1 mg/ml; Roche) in RPMI+ medium. Single cell suspensions were prepared by filtration through a 70-µm filter. Red blood cells were lysed using ammonium chloride (Stem Cell Technologies, Grenoble, France). Single cells were plated into a 25-cm (*Parsons et al., 2012*) culture flask. After 1–3 weeks, when cells reached 60–70% confluency, fibroblasts were removed using mouse anti-human CD90 (Clone AS02; Dianova, Hamburg, Germany) bound to Mouse Pan IgG Dynabeads (Life Technologies, Erembodegem, Belgium). Cell cultures were subsequently passaged at 70–90% confluency and banked at −80°C. Primary tumor cell cultures were grown in RPMI Medium 1640 supplemented with 20% fetal bovine serum (FBS), 2 mM L-Glutamine, 100U/ml penicillin, 100 µg/ml streptomycin, 1 µg/ml fungizone, and 10 µg/ml gentamicin (Life Technologies) up to 25 passages.

## Immunofluorescent double staining for γH2AX and RAD51

Cells were seeded in 8-well Lab-tek Permanox Chamber slides (Nunc, Zellik, Belgium), treated for 24 hr, fixed in 4% paraformaldehyde for 15 min at room temperature, and ice-cold methanol for 5 min. Primary antibodies recognizing γH2AX (JBW301, Millipore, Overijse, Belgium) and RAD51 (PC130, Merck, Darmstadt, Germany) followed by secondary antibodies conjugated to Alexa Fluor 647 and 488 (Life Technologies) were used. Images were acquired using an A1R Eclipse Ti inverted confocal microscope (Nikon, Brussels, Belgium) and processed using Fiji software, with compound or vehicle-treated cells being processed identically. Nuclei with >5 foci were scored as positive, and at least 200 nuclei were counted per condition by two independent individuals, blinded to the genotypes.

## Cell cycle analysis with BrdU and propidium iodide

Cells were treated for 24 hr with 26 µM olaparib, 0.3 µM mitomycin C, 0.03 µM camptothecin or carrier, and incubated for 90 min with BrdU (10 µM) before harvesting. Cells were resuspended in ice-cold PBS and ice-cold ethanol was slowly added to 70%. Cells were fixed for 5 min at room temperature, treated with 2 M HCl for 30 min and stained with FITC-conjugated anti-BrdU antibody (BD). Cells were washed, resuspended in PI/RNase staining buffer (BD), and analyzed on a BD Biosciences FACSVerse flow cytometer. Cell cycle distributions were modeled using FlowJo software, and the fraction of cells in S-phase, G2/M and G1 determined as described by *Watson et al. (1987)*.

## Cytotoxicity screening

5,000 cells/well were seeded in 96-well plates. After 24 hr, cells were treated in quadruplicate, incubated for 48 hr at 37°C and analyzed using the In Vitro Toxicology Assay Kit, Sulforhodamine B-based (Sigma, Diegem, Belgium) as per the manufacturer's instructions. Growth inhibition was calculated as described (*Vichai and Kirtikara, 2006*).

## siRNA knockdown

siRNA ON-TARGETplus SMART pools (Thermo) were diluted in Optimem I reduced serum medium using Lipofectamine RNAiMAX (Life technologies) to reverse transfect MCF7 cells For cytotoxicity

screening, transfections were in 96-well format and medium was changed 14 hr after transfection. Cells were treated with olaparib (26 µM) and after 48 hr processed for cytotoxicity screening. Simultaneously, siRNA transfections in 12-well plates were done to quantify knockdown.

## Gene expression

Total RNA was extracted using the RNeasy Mini kit (Qiagen, Venlo, The Netherlands) and reverse transcribed using the SuperScript III reverse transcription system (Life technologies). Quantitative RT-PCR (qRT-PCR) with *ACTB* an internal control was performed using TaqMan gene expression assay probes and 5 µl TaqMan Fast Universal PCR master mix (Life technologies). Reactions were amplified in a Roche LightCycler 480 using the following cycles: 50°C (2 min), 95°C (30 s), and 40 cycles of 95°C (3 s), 60°C (30 s).

## Antibodies, compounds, and other reagents

Mouse anti-phospho-Histone H2A.X (Ser139) monoclonal antibody (clone JBW301) was from Millipore Corporation, Billerica, MA, USA. Rabbit anti-Rad51 (PC130) polyclonal antibody was from Calbiochem/ Merck, Darmstadt, Germany. Rabbit anti-ACTB (#4967) polyclonal antibody was from Cell Signalling, Danvers, MA, USA. FITC-conjugated anti-BrdU antibody (347583) was from Becton–Dickinson, San Jose, CA, USA. Alexa Fluor 647 goat anti-mouse IgG (A-21235) and Alexa Fluor 488 goat anti-rabbit IgG (A-11034) were from Life technologies, Carlsbad, CA, USA. Olaparib (AZD-2281, batch JSAR104) was purchased from JS Research Chemicals Trading, Schleswig Holstein, Germany. Cis-platinum (II) diammine dichloride (P4394), paclitaxel (T7402), mitomycin C (M4287), (S)-(+)-camptothecin (C9911) and carmustine (C0400) were purchased from Sigma-Aldrich, St. Louis, MO, USA, and prepared and stored according to the manufacturer's recommendations. siRNA ON-TARGETplus SMART pools were purchased from Thermo Scientific Dharmacon, Chicago, IL, USA: Non-targeting (D-001810-10-05); ATM (L-003201-00-0005); ATR (L-003202-00-0005); BRCA1 (L-003461-00-0005); and BRCA2 (L-003462-00-0005). TaqMan gene expression assays (Life technologies, Carlsbad, CA, USA) used in this study were as follows ATM: Hs01112355_g1; ATR: Hs00992123_m1; BRCA1: Hs01556193_m1; BRCA2: Hs00609073_m1; ACTB: Hs99999903_m1. Normal goat serum (005-000-121) was from Jackson Immunoresearch Labs, West Grove, PA USA.

## Acknowlegements

We greatly appreciate the assistance of Mark Veugelers and Stéphane Plaisance of the VIB Technology Watch team. We acknowledge the contributions of Gilian Peuteman and Thomas Van Brussel for Sequenom validation experiments. We thank Penelope Webb, Daniel Buchanan, Kaltin Ferguson, Mike Walsh, Joanne Young, as well as ANECS collaborators, ANECS staff and participating Institutions (http://www.anecs.org.au/html) for their roles in ANECS study setup and/or characterization of ANECS endometrial tumors. We are grateful to the Verelst Fund and Reliable Cancer Therapies. The research was funded by grants from the Fund for Scientific Research Flanders (FWO-F), the 'Stichting tegen Kanker' and the KULeuven (KUL PFV/10/016 SymBioSys). ANECS patient recruitment, data collection, biospecimen collection, and IHC analysis was supported by funding from the National Health and Medical Research Council (NHMRC) of Australia (Grant ID#339435); The Cancer Council Queensland (ID#4196615); and Cancer Council Tasmania (IDs#403031, #457636), and Cancer Australia (ID1010859). HZ, BT, LC, and JR hold a FWO postdoctoral fellowship, BTY and MM hold a FWO PhD fellowship. AS is supported by an NHMRC Senior Research Fellowship.

## Additional information

### Competing interests

DL, an inventor on a patent application regarding the use of recurrent indels to detect MSI. The VIB is owner of this patent application, and the said patent application has been licensed to an outside company. Neither VIB nor any of the authors have equity stakes in the company. However, VIB stands to eventually receive royalties. The other authors declare that no competing interests exist.

### Funding

| Funder | Grant reference number | Author |
|---|---|---|
| Stichting tegen Kanker | ZKC6069 | Diether Lambrechts |

| Funder | Grant reference number | Author |
|---|---|---|
| KU Leuven | PFV/10/016 SymBioSys | Diether Lambrechts |
| National Health and Medical Research Council | #339435 | Amanda Spurdle |
| Cancer Council Queensland | #4196615 | Amanda Spurdle |
| Cancer Council Tasmania | #403031, #457636 | Amanda Spurdle |
| Cancer Australia | 1010859 | Amanda Spurdle |
| Fonds Wetenschappelijk Onderzoek | Postdoctoral Fellowship | Hui Zhao, Bernard Thienpont, Lieve Coenegrachts |
| Fonds Wetenschappelijk Onderzoek | PhD fellowship | Betül Tuba Yesilyurt, Matthieu Moisse |
| Fonds Wetenschappelijk Onderzoek | G.0772.13N | Diether Lambrechts |
| National Health and Medical Research Council | Senior Research Fellowship | Amanda Spurdle |

The funders had no role in study design, data collection and interpretation, or the decision to submit the work for publication.

## Author contributions

HZ, DL, Conception and design, Acquisition of data, Analysis and interpretation of data, Drafting or revising the article, Contributed unpublished essential data or reagents; BT, JR, Conception and design, Acquisition of data, Analysis and interpretation of data, Drafting or revising the article; BTY, MM, XS, AS, Conception and design, Acquisition of data, Analysis and interpretation of data; LC, GM, JC, Conception and design, Acquisition of data; SS, DS, SA, AM, Acquisition of data, Analysis and interpretation of data; FA, Conception and design, Acquisition of data, Analysis and interpretation of data, Contributed unpublished essential data or reagents

## Author ORCIDs

Bernard Thienpont, http://orcid.org/0000-0002-8772-6845
Matthieu Moisse, http://orcid.org/0000-0001-8880-9311

## Ethics

Human subjects: Informed consent and consent to publish was obtained from all patients. Ethical approval was obtained at the ethical committee of University Hospital Gasthuisberg of Leuven with identifier ML2266.

# Additional files

## Supplementary file

• Supplementary file 1.

## Major datasets

The following datasets were generated:

| Author(s) | Year | Dataset title | Dataset ID and/or URL | Database, license, and accessibility information |
|---|---|---|---|---|
| Zhao Hui | 2013 | Signatures of mismatch repair deficiency in cancer genomes | https://www.ebi.ac.uk/ega/studies/EGAS00001000182 | Access to datasets must be approved by the specified Data Access Committee (DAC). |
| Zhao Hui | 2012 | Complete Genomics paired end sequencing; Ovarian cancer | https://www.ebi.ac.uk/ega/studies/EGAS00001000158 | Access to datasets must be approved by the specified Data Access Committee (DAC). |

The following previously published dataset was used:

| Author(s) | Year | Dataset title | Dataset ID and/or URL | Database, license, and accessibility information |
|---|---|---|---|---|
| The Cancer Genome Atlas Network | 2013 | Data from: Comprehensive molecular characterization of human colon and rectal cancer | http://dx.doi.org/10.1038/nature11252 | Available for download as a supplementary file. |

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
