## [Decision Letter]

Thank you for sending your work entitled “Mismatch repair deficiency endows tumors with a unique mutation signature and sensitivity to DNA double-strand breaks” for consideration at *eLife.* Your article has been favorably evaluated by Stylianos Antonarakis (Senior editor), a Reviewing editor, and 2 reviewers, one of whom, Thilo Dörk, has agreed to reveal his identity.

The Senior editor has assembled the following comments to help you prepare a revised submission.

1) There is a major issue of mapping and calling the in-dels within short repeats that are typical of MSI + cancers. This can lead to unmapped reads, mismapping and type 1 and type 2 errors. This problem especially affects the gapped reads in the Complete Genomics platform. Strategies to mitigate these issues are not mentioned. I have considerable doubts about the inclusion of the whole-genome data in this manuscript - after all, N = 3 is a very small number anyway – and I suggest that the issue should be addressed in exome data (even if some of those data are from the WGS cancers).

2) The sample set is heterogeneous in terms of cancer of origin and derivation from primary cancers and cultures that are likely to have been subjected to considerable in vitro selection pressure and/or founder effects.

3) The similarity between germline and somatic mutation spectra might, in part be caused by many somatic mutations occurring in normal progenitors prior to loss of MMR. Is there a way of investigating this in comparison with non-MMR tumors (e.g. examining effects of age)?

4) The 59-marker exomic MSI panel is useful, especially for Lynch syndrome, and appears to perform well.

5) The pathway analysis presumably incorporated all detected variants. Whilst strongly suggestive, does filtering for variants with strong evidence of functionality alter these conclusions? Moreover, the burden of mutations in these pathways might relate to redundant function – noting that MSI + cancers are usually near-diploid - rather than positive selection. This may or may not matter for therapeutic purposes, but can it be checked in some way?

---

## [Author Response]

1) There is a major issue of mapping and calling the in-dels within short repeats that are typical of MSI + cancers. This can lead to unmapped reads, mismapping and type 1 and type 2 errors. This problem especially affects the gapped reads in the Complete Genomics platform. Strategies to mitigate these issues are not mentioned. I have considerable doubts about the inclusion of the whole-genome data in this manuscript - after all, N=3 is a very small number anyway – and I suggest that the issue should be addressed in exome data (even if some of those data are from the WGS cancers).

We thank the reviewer for this comment. Indeed, different sequencing platforms are each characterized by their specific false-positive and false-negative variant detection rates. For example, 10.5% of indels was false-positive in Complete Genomics (CG) datasets, whereas 27.7% of indels was missed (false-negatives), as discussed by Zook et al (25). For Illumina genomes, 6.9% of indels were false-positive and 0.5% of indels were false-negative.

First, to address the issue of false-positives (type 1 error), we expanded the number of randomly selected indels from the CG-sequenced MMR-deficient tumor (MMR-1) and Illumina-sequenced MMR-deficient tumor (MMR-2), respectively. We chose an orthogonal validation technology, i.e., Sequenom MassARRAY, to validate a total number of 391 indels. The overall validation rate that we obtained for indels was 90.3% in CG genomes and 85.9% in Illumina -sequenced genomes (see Table 1 below and [Supplementary-material SD3-data]). These validation rates are thus very similar between both platforms. The validation rates are also much higher than those that we observed for indels in MMR-proficient tumors and those the reviewer is correctly referring to in the literature (44; 45). The low validation rates for indels in the MMR-proficient whole-genome tumors most probably reflect the fact that in germ-line genomes, as well as MMR -proficient tumor genomes, the number of true-positive indels is low in comparison to the number of false-positive indels that are detected. However, in MMR-deficient tumors, due to their specific hypermutator phenotype, the number of true-positive indels is vastly increased, thereby rendering the false positive fraction proportionally much smaller. In the revised manuscript, we have therefore highlighted the possibility of having false-positive and false-negative findings in the whole-genomes and indicate that this may affect observed indel rates. We also discuss that validation rates in MMR-deficient tumors are much higher than in the MMR-proficient tumors and explain the reasons for this. As highlighted by the reviewer, sequence reads containing both indels and substitutions (i.e., reads with a relatively high percentage of mismatches) are more prone to mismapping than sequence reads containing only substitutions or indels. We acknowledge this issue, and have in response deleted the paragraph describing that somatic indels and substitutions are often located close to each other in the 3 MMR-deficient tumors that were whole-genome sequenced. Indeed, such observations are prone to contamination by false-positives.

Table 1.Validation rate of somatic indels detected upon whole-genome sequencing. These additional validation experiments for indels have been included in the revised manuscriptTumorSomatic IndelsConfirmedNot confirmedValidation rateMMR- 11872090.3%MMR- 21032183.1%MMR- 354690.0%MMR+ 1090.0%MMR+ 22722.2%

Secondly, we have followed the reviewer’s suggestion and have removed the 3 whole-genomes from the analyses aimed at identifying recurrent mutations and constructing a novel MSI panel. In particular, we repeated all analyses with only the 13 genomes subjected to Illumina exome-sequencing. We observed that in coding regions, 1.4% of homopolymers were affected at least once (i.e. in 2073 homopolymers out of a total of 29,663), whereas 414 were affected at least twice. Furthermore, 47 homopolymers were affected in ≥5 samples. In 3’UTR and 5’UTRs, 2296 and 105 homopolymers were affected in ≥5 samples respectively. When randomly selecting recurrent indels to design a panel of recurrent markers capable of assessing MSI, 54 out of the 59 originally selected markers were still selected, as they affected ≥5 out of 13 tumors (compared to 59 recurrent indels affecting ≥6 out of 16 tumors). Of these 54 markers, 45 markers were in UTRs and 9 were in coding regions. Applying the 54-marker panel on the same discovery set of 236 EM tumors as described in the original manuscript, we also found 3 positive markers as the threshold with the best Matthew Correlation Coefficient (Figure 8).

When comparing our 54 markers against the Bethesda panel, we equally found that the 54-maker panel had a higher sensitivity compared to Bethesda. Specifically, we applied the 54 -marker panel to a set of 114 endometrial tumors and a set of 126 colorectal tumors as described in the original manuscript. For the EM tumors, 73 tumors (64%) were defined as MSS/MSI-L and 41 tumors (36%) as MSI-H. Out of these 41 MSI-H tumors, Bethesda identified 29 tumors as MSI-H (>2 markers positive), 7 tumors as MSI-L and 5 tumors as MSS. Vice versa, Bethesda did not identify any MSI-H tumor that was not identified by our novel MSI panel (Figure 8). IHC on 9 out of 12 discordant samples confirmed that each of these samples was deficient either for MLH1 or MSH2, and thus MMR-deficient. No tumor slides were available for the remaining 3 samples. The 9 discordant samples, we had access to, were confirmed as true positives by IHC. For CRC tumors, there were 97 MSS tumors in our 54-marker panel that were concordantly called MSS or MSI-L by the Bethesda panel. The remaining 29 samples were detected as MSI in the 54-marker panel (Figure 8). 28 of these were also called MSI-H by the Bethesda panel, whereas one was called MSS by the Bethesda panel. It had a *BRAF* mutation and was *MLH1* hypermethylated, thereby confirming MMR-deficiency and correct classification by the 54-marker panel.

Finally, we also repeated our pathway analyses on the genes affected by indels in the 13 exomes, rather than on the whole set of 16 exomes, 3 of which were generated by whole-genome sequencing. Pathway analyses on the 3856 genes affected by a somatic indel using IPA^®^ revealed that the “*Role of BRCA1 in DNA damage response*” was the top enriched pathway (*P*=4.2E-04). IPA^®^ analysis of 1302 genes affected by recurrent indels revealed that the *“DNA double-strand break (DSB) repair by Homologous Recombination (HR)”* was the top enriched pathways (*P*=4.7E-03). Pathway analyses of 6736 indels in MMR-deficient tumors using GenomeMuSiC revealed that the “*ATR/BRCA pathway*”, “*Homologous recombination repair*” and “*DNA repair*” pathways were ranked highest in BioCarta, DNARepairDB and Reactome databases respectively (*P*=4.9E-13, *P*=1.8E-03and*P*=5.7E-08, respectively). Overall, these results are nearly identical to the data generated on the 16 exomes, as presented in the original manuscript.

In conclusion, since our data were not significantly affected or did not change any of our conclusions, depending on whether we analyzed 13 or 16 genomes, we chose to present the data of the 16 exomes as the main analysis. However, in the revised manuscript, we have now added a sentence highlighting that data and conclusions did not change when the analysis was limited to the 13 exomes generated by Illumina exome-sequencing only. Furthermore, since the response to this comment will be published in parallel to the manuscript, a critical reader will be able to assess in full detail that data did not change after removing the 3 whole-genomes from the analysis.Author response image 1.The 54-marker panel generated from 13 Illumina-sequenced exomes for MSI testing.

*2) The sample set is heterogeneous in terms of cancer of origin and derivation from primary cancers and cultures that are likely to have been subjected to considerable* in vitro *selection pressure and/or founder effects*.

We agree with the reviewer that it is important to consider the heterogeneity of the tumors in terms of cancer of origin or derivation procedure. We have therefore performed a clustering analysis of all MMR-deficient tumors that we sequenced for the genes affected by either a somatic substitution or indel in the coding region. As can be appreciated from Figure 9 below, no obvious subgroup in terms of cancer of origin was observed. In the revised manuscript, this figure has also been added as Figure 4—figure supplement 1.Author response image 2.Clustering analysis of all samples based on the genes carrying somatic mutations in their coding regions.

The figure also shows that there is no distinct difference between primary tumors and data generated on the primary cell cultures. As mentioned in the revised manuscript, we specifically chose to use primary tumor cultures of low passage rather than tumor cell lines, because the latter have been subject to much more selective pressure as primary tumor cultures of low passage. In addition to the above cluster analysis, we also performed pathway analysis on the 10 primary tumor tissues only. Pathway analyses of all genes affected by a somatic indel using IPA^®^ revealed that the “*Role of BRCA1 in DNA damage response*” and “*DNA double-strand break (DSB) repair by Homologous Recombination (HR)*” were the top enriched pathways (*P*=6.5E-03 and *P*=1.1E −02, respectively). IPA^®^ analysis of genes affected by recurrent indels revealed that the “*Role of BRCA1 in DNA damage response*” was also enriched (p = 2.0E-03). Pathway analyses of all indels in MMR-deficient tumors using GenomeMuSiC revealed that the “*ATR/BRCA pathway*”, “*Homologous recombination repair*” and “*DNA repair*” pathways were ranked highest in BioCarta, DNARepairDB and Reactome databases respectively (*P*=1.0E -09, *P*=0.4E-02and *P=*3.4E-06, respectively). The results derived only from the primary tumors suggest that MMR-deficient tumors are indeed enriched in indels affecting the DSB repair pathway. Data are thus highly concordant with the results shown in the manuscript.

*3) The similarity between germline and somatic mutation spectra might, in part be caused by many somatic mutations occurring in normal progenitors prior to loss of MMR. Is there a way of investigating this in comparison with non-MMR tumors (e.g*. *examining effects of age)?*

We thank the reviewer for this insightful hypothesis. However, we found no correlation between age at diagnosis and the number of mutations detectable (*P*=0.86). Moreover, although the age at diagnosis of patients with MMR-proficient and MMR-deficient tumors was very comparable (67 and 62 years respectively), MMR-deficient tumors carried >55 times more mutations than MMR-proficient tumors. When compared to MMR- deficient tumors, MMR-proficient somatic mutations thus comprise at most only 2% of all mutations in MMR-deficient tumors, a fraction that is unlikely to contribute significantly to the similarity in patterns between MMR -deficient and germline mutation patterns.

Finally, as described in the original manuscript (Figure 2 in the original manuscript), no extensive similarity was noted between MMR-proficient and germline mutation patterns. Consequently, even if somatic mutations (as reflected in MMR-proficient mutations) would contribute significantly to MMR-deficient mutation patterns, they would not display extensive similarity to germline variation patterns and could therefore not be responsible for the patterns observed in MMR - deficient tumors. In the revised manuscript, we have indicated this.

*4) The 59-marker exomic MSI panel is useful, especially for Lynch syndrome, and appears to perform well*.

We are happy to read that the reviewers appreciate our work.

*5) The pathway analysis presumably incorporated all detected variants. Whilst strongly suggestive, does filtering for variants with strong evidence of functionality alter these conclusions? Moreover, the burden of mutations in these pathways might relate to redundant function – noting that MSI + cancers are usually near-diploid - rather than positive selection*. *This may or may not matter for therapeutic purposes, but can it be checked in some way?*

We apologize for not more clearly explaining our pathway analyses in the original manuscript. We described two types of pathway analyses: the first involved somatic frameshift indels in exons, the second involved somatic indels both in exons and exon/intron boundaries. We thus already restricted the presented pathway analyses to variants with strong evidence of functionality. Indeed, in the first analysis each of the selected somatic indels in exons already represented an out-of-frame mutation, thus conferring a heterozygous loss-of function mutation on the gene affected in the tumor.

In an effort to further enrich for mutations with a functional effect in the tumor, we additionally restricted our pathway analyses to genes expressed in endometrial or colorectal normal tissue. RNA-sequencing data generated on normal endometrium and colon tissue were downloaded from TCGA (47; 40). For both EM and CRC datasets, we calculated the mean normalized read count for each gene in 12 normal endometrial samples and 40 normal colorectal samples respectively. Transcripts with over 10 reads per kb and per million reads were considered expressed. This resulted in 12,851 and 12,293 genes that were expressed in endometrial and colorectal tissues respectively. We then limited the pathway analyses to indels affecting genes expressed either in normal endometrium or in normal colon tissue. Pathway analysis using IPA^®^ of 2,126 expressed genes affected by a somatic indel ranked the “*Role of BRCA1 in DNA damage response*” as the top enriched pathway. IPA^®^ analysis of 851 expressed genes affected by recurrent indels revealed that the “*Double-strand break repair by homologous recombination*” pathway ranked top. GenomeMuSiC ranked “*ATR/BRCA pathway*”, “*DNA repair*” and “*Homologous recombination*” pathways as the top pathways for BioCarta, Reactome and DNARepair DB respectively. By restricting ourselves to frameshift indels affecting genes that are expressed in endometrial tissue, similar results were thus obtained.

In order not to burden the reader with too many pathway analyses, we have chosen not to present these data in the revised manuscript. Furthermore, since the response to this comment will be published online, critical readers will be able to appreciate in detail that the outcome was not altered by removing genes that are affected by indels but not expressed in the normal corresponding tissue.

**References**

1. Zook, J. M. et al. Integrating human sequence data sets provides a resource of benchmark SNP and indel genotype calls. Nat Biotechnol 32, 246‐251, doi:10.1038/nbt.2835 (2014).

2. Jia, P. et al. Consensus rules in variant detection from next-‐generation sequencing data. PLoS One 7, e38470, doi:10.1371/journal.pone.0038470 (2012).

3. O'Rawe, J. et al. Low concordance of multiple variant-‐calling pipelines: practical implications for exome and genome sequencing. Genome medicine 5, 28, doi:10.1186/gm432 (2013).

4. TCGA. Comprehensive molecular characterization of human colon and rectal cancer. Nature 487, 330‐337, doi:10.1038/nature11252 (2012).

5. Cancer Genome Atlas Research, N. et al. Integrated genomic characterization of endometrial carcinoma. Nature 497, 67‐73, doi:10.1038/nature12113 (2013).